# Unsupervised Hierarchical Skill Discovery

**Damion Harvey** [1]  **Geraud Nangue Tasse** [1 2]  **Benjamin Rosman** [1 2]  **Branden Ingram** [1 2]  **Steven James** [1 2]

## Abstract

We consider the problem of unsupervised skill segmentation and hierarchical structure discovery in reinforcement learning. While recent approaches have sought to segment trajectories into reusable skills or options, most rely on action labels, rewards, or handcrafted annotations, limiting their applicability. We propose a method that segments unlabelled trajectories into skills and induces a hierarchical structure over them using a grammar-based approach. The resulting hierarchy captures both low-level behaviours and their composition into higher-level skills. We evaluate our approach in high-dimensional, pixel-based environments, including Craftax and the full, unmodified version of Minecraft. Using metrics for skill segmentation, reuse, and hierarchy quality, we find that our method consistently produces more structured and semantically meaningful hierarchies than existing baselines. Furthermore, as a proof of concept, we demonstrate that these discovered hierarchies accelerate and stabilise learning on downstream reinforcement learning tasks.

## 1. Introduction

Human planning operates hierarchically, reasoning in terms of goals and sub-tasks rather than primitive actions (Correa et al., 2025; Ho et al., 2019). Similarly, reinforcement learning (RL) agents in high-dimensional environments like Minecraft benefit from hierarchical decompositions to improve learning efficiency and policy reuse (Tessler et al., 2017; Nachum et al., 2019). While manually defined hierarchies, such as Hierarchical Task Networks (HTNs) (Erol et al., 1994), are effective, they demand significant human effort and domain expertise to create (Chen et al., 2021).

[1]University of the Witwatersrand, Johannesburg, South Africa [2]Machine Intelligence and Neural Discovery (MIND) Institute, University of the Witwatersrand, Johannesburg, South Africa. Correspondence to: Damion Harvey <damion.harvey1@students.wits.ac.za>.

*Proceedings of the $43^{rd}$ International Conference on Machine Learning*, Seoul, South Korea. PMLR 306, 2026. Copyright 2026 by the author(s).

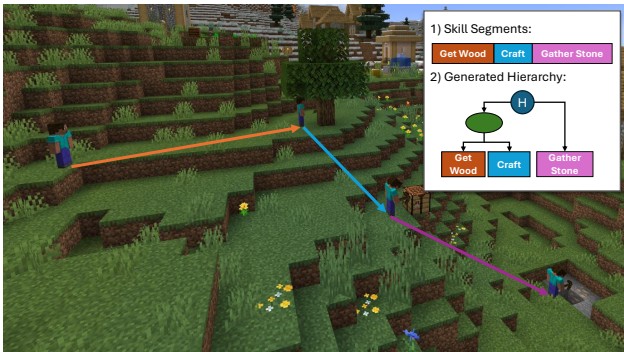

*Figure 1.* Example of HiSD applied to a Minecraft trajectory. **Step 1**: HiSD segments the observational trajectory into distinct skills, such as get wood, craft tools, and gather stone. **Step 2**: Using these segmented trajectories, HiSD applies a grammar-based compression algorithm to induce a hierarchy over the discovered skills, revealing reusable subroutines and their temporal organisation.

An alternative is learning structure directly from data. Prior approaches explore this via reward signals or action supervision (Ranchod et al., 2015; Kipf et al., 2019); however, these methods rely on strong assumptions, such as access to action labels or known skill orderings, and typically produce flat segmentations rather than deep, compositional hierarchies.

We propose Hierarchical Skill Discovery (HiSD), a fully unsupervised framework for extracting reusable, multi-level skill hierarchies purely from observational data. Crucially, HiSD decouples structure discovery from policy execution: unlike baselines that require actions during discovery (Lu et al., 2021; Kipf et al., 2019), HiSD operates on observations only, enabling scalability to abundant unlabelled video data (Baker et al., 2022). Our method combines temporal action segmentation (TAS) to identify latent skills based on visual coherence, and grammar-based sequence compression to induce structured hierarchies over them. An overview of this method is pictured in Figure 1.

We evaluate HiSD on Craftax (Matthews et al., 2024) and Minecraft (Baker et al., 2022), two domains requiring complex sequential decision-making. Across these domains, HiSD delivers the strongest overall performance among the compared methods, producing accurate skill segmentations and more coherent multi-level hierarchies with fewer assumptions. Our contributions are: (1) a method integrating TAS and grammar induction for unsupervised structure dis-

covery; (2) an observational feature-only pipeline suitable for unlabelled demonstrations; (3) empirical validation of superior segmentation quality and abstraction depth; and (4) a proof of concept demonstrating the utility of these representations in accelerating downstream reinforcement learning.[1]

## 2. Background

### 2.1. Skill Segmentation and Identification

Temporal action segmentation aims to partition a continuous observation sequence $T = (X_1, \ldots, X_n)$ into $M$ temporally contiguous segments, where $X \in \mathbb{R}^d$ is a feature vector, and $n$ is the length of the sequence, assigning each a discrete skill label $z_t \in \{1, \ldots, K\}$. Here $K$ represents the maximum number of skills in the dataset. In the unsupervised setting, neither the ground truth labels nor the boundaries are known.

Recent state-of-the-art approaches formulate this as an optimal transport problem. Specifically, ASOT (Xu & Gould, 2024) relaxes the segmentation into a soft assignment problem. Let $C \in \mathbb{R}^{n \times K}$ be a cost matrix whose entry $C_{tk}$ encodes the visual dissimilarity between observation feature $X_t$ and the $k$-th latent skill prototype. ASOT solves for an assignment plan $\Gamma \in \mathbb{R}_+^{n \times K}$ by minimising a regularised objective:

$$\min_{\Gamma} \quad \langle C, \Gamma \rangle + \alpha \mathcal{R}_{\text{temp}}(\Gamma) + \lambda D_{\text{KL}}(\Gamma^\top \mathbf{1}_n \,\|\, q), \quad (1)$$

where $\alpha \in [0, 1]$ weights the temporal regularity term against the visual matching cost, $\lambda > 0$ controls the strength of the aggregate-skill marginal penalty, $\mathbf{1}_n \in \mathbb{R}^n$ is the all-ones vector (so that $\Gamma^\top \mathbf{1}_n$ is the column-marginal of $\Gamma$, i.e. the aggregate distribution over skills induced by the assignment), and $q \in \Delta_K$ is a target prior (typically uniform) over the $K$ latent skills.

The temporal regularity term $\mathcal{R}_{\text{temp}}$ is realised as a Gromov-Wasserstein (GW) component comparing two intra-space cost matrices: a frame-side matrix $C^v \in \mathbb{R}^{n \times n}$ encoding temporal proximity, and a skill-side matrix $C^a \in \mathbb{R}^{K \times K}$ encoding skill identity. With radius parameter $r \in [0, 1]$, these are defined element-wise as

$$C_{ik}^v = \begin{cases} 1/r, & 1 \leq |i - k| \leq nr \\ 0, & \text{otherwise} \end{cases}, \quad C_{jl}^a = \mathbf{1}[j \neq l], \quad (2)$$

and combined under the quadratic GW loss $L(a, b) = ab$ to yield

$$\mathcal{R}_{\text{temp}}(\Gamma) = \sum_{\substack{i,k \in [n] \\ j,l \in [K]}} L\big(C_{ik}^v, C_{jl}^a\big) \Gamma_{ij} \, \Gamma_{kl}. \quad (3)$$

Intuitively, $\mathcal{R}_{\text{temp}}$ penalises assignments in which two frames within $nr$ steps of each other are mapped to different skills, while imposing no penalty on transitions outside this temporal radius, nor on adjacent frames mapped to the same skill. The radius $r$ therefore directly controls the minimum expected segment length, enforcing local smoothness and preventing rapid flickering of labels.

This formulation utilises unbalanced optimal transport, which allows the algorithm to handle missing and repeated skills. It does not force every latent skill prototype to appear in every episode, nor does it enforce a specific sequential ordering. Furthermore, the KL-divergence term ($D_{\text{KL}}$) acts as a soft constraint on the aggregate skill distribution. This enables the model to handle unbalanced datasets, where certain skills dominate the dataset while critical interaction skills appear only briefly, a characteristic of the datasets used in this work (Xu & Gould, 2024).

### 2.2. Grammar Sequence Compression

To represent hierarchical structure, we employ the formalism of context-free grammars (CFGs). A CFG is defined as a tuple $G = (\mathcal{N}, \Sigma, \mathcal{P}, S_0)$, where $\Sigma$ is a set of terminal symbols (atomic units), $\mathcal{N}$ is a set of non-terminal symbols (abstract variables), $\mathcal{P}$ is a set of production rules replacing non-terminals with sequences of symbols, and $S_0$ is the distinct start symbol.

A prominent method for grammar induction is Sequitur (Nevill-Manning & Witten, 1997), a linear-time algorithm capable of inferring hierarchical structure from data. Sequitur operates on a single discrete input string of terminal symbols in $\Sigma$, and incrementally builds a hierarchy to compress it. It maintains two invariants: **(1) Digram Uniqueness:** No pair of adjacent symbols appears more than once in the grammar. If a repetition is found, a new non-terminal rule is created to replace it. **(2) Rule Utility:** Every rule must be used at least twice. Rules used only once are removed and their contents expanded. Sequitur processes the input string deterministically, producing a grammar where the start symbol $S_0$ expands to exactly reproduce the original input string. The resulting derivation tree forms a hierarchy: the root is the full trajectory, internal nodes are discovered non-terminal subroutines, and leaves are the atomic terminal skills. This is because non-terminals can consist of both other non-terminals and terminals, allowing a recursive structure.

## 3. Related Work

Long-horizon decision-making benefits from temporal abstraction and skill reuse, which allow agents to operate at multiple levels of planning (Correa et al., 2025; Sutton et al., 1999; Solway et al., 2014). We focus on the offline,

---

[1]All code used is available on our `GitHub Repository`.

observation-only setting, where trajectories contain state features but no actions, rewards, or prior segmentation (Lu et al., 2021).

### 3.1. Skill Segmentation and Discovery

Skill segmentation typically assumes access to state-action trajectories or rewards (Ranchod et al., 2015; Kipf et al., 2019). Approaches such as Compositional Imitation Learning and Execution (CompILE) (Kipf et al., 2019) and Option-Critic (Bacon et al., 2017) segment demonstrations into latent skills or options to accelerate RL. However, these methods generally rely on action supervision. CompILE, in particular, suffers from producing location-centric, redundant skills and necessitates prior knowledge of the average length per skill per trajectory (and the number of segments). Similarly, Bayesian methods like Nonparametric Bayesian Reward Segmentation (NPBRS) (Ranchod et al., 2015) are capable of inferring the number of skills from data but typically require heavy supervision, such as explicit rewards, action labels, and environment interaction.

Other approaches, including spectral or change-point methods (Zhu et al., 2022; Konidaris et al., 2012) and online unsupervised methods (Eysenbach et al., 2019), typically require active environment interaction. More recent work such as SloTTAr (Gopalakrishnan et al., 2023) extends CompILE and the Ordered Memory Policy Network (OMPN) (Lu et al., 2021) with slot-based transformers to handle variable subroutine counts. However, it still operates on state-action trajectories to reconstruct action sequences and produces flat segmentations rather than multi-level hierarchies. In contrast, we discover skills from pre-collected observations without access to actions, rewards, or interaction.

### 3.2. Learning Multi-level Hierarchies from Demonstration

Several methods aim to learn multi-level task hierarchies, commonly formalised as HTNs (Erol et al., 1994). These approaches can be broadly divided into those that rely on structured symbolic input and those that operate under weak supervision. Structured-input methods assume substantial prior knowledge in the form of annotated plans, logical schemas, or explicit skill decompositions. For example, HTN-Maker (Hogg et al., 2008), Circuit-HTN (Chen et al., 2021), and CurricuLAMA (Nejati et al., 2006) induce hierarchical structures from heavily annotated data, requiring domain expertise to label and segment tasks, as well as access to correct skill sequencing.

In contrast, weakly supervised approaches attempt to infer hierarchical structure directly from demonstrations using some form of labels. Clique-Chain HTN (Hayes & Scassellati, 2016) and OMPN learn hierarchical structure from demonstrations; however, OMPN still depends on action

labels, known skill orderings at inference time, and a predefined hierarchy depth. Grammar-based methods (Lange & Faisal, 2019) similarly extract hierarchical macro-actions from sequences of primitive actions.

## 4. Hierarchical Skill Discovery

The fundamental challenge in learning from demonstration lies in parsing continuous, unlabelled observation streams into distinct, reusable behavioural units. While this objective is shared by prior frameworks such as NPBRS (Ranchod et al., 2015), CompILE (Kipf et al., 2019), and OMPN (Lu et al., 2021), we seek to achieve this without relying on action labels, reward signals or online interaction. To this end, we propose Hierarchical Skill Discovery, a framework to extract reusable, multi-level skill hierarchies directly from raw observational features. Our method bridges the gap between low-level continuous control and symbolic planning by treating skill discovery as a two-stage process: (1) *Segmentation*, which discretises continuous dynamics into atomic behavioural units, and (2) *Structure Induction*, which compresses these units into a compositional grammar. This framework is presented in Figure 2.

### 4.1. Unsupervised Skill Identification

Given a dataset of $N$ unlabelled observation trajectories $\{T^{(1)}, \ldots, T^{(N)}\}$, we first aim to convert continuous features into discrete skill sequences, as seen in Figure 2a. The grammar-induction stage in Section 4.2 operates on any sequence of discrete skill indices and is therefore fully decoupled from the choice of segmentation algorithm; HiSD is compatible with any TAS method that produces such labels. For our experiments, we instantiate this stage using the ASOT framework described in Section 2.1. We treat the entire dataset of trajectories as a batch. For a given trajectory $T^{(i)}$ of feature embeddings $X^{(i)}$, we compute the optimal transport plan $\Gamma^*$ by solving Equation 1. We obtain frame-level discrete skill indices $z_{1:n}^{(i)}$, with $z_t^{(i)} \in \{1, \ldots, K\}$, via a hardening step $z_t^{(i)} = \arg\max_{k \in \{1,\ldots,K\}} \Gamma_{tk}^*$. These are integer indices representing the discovered skills. Crucially, unlike clustering methods that ignore time, this approach enforces temporal consistency, ensuring that $z$ remains constant over coherent segments of behaviour. This effectively transforms the continuous trajectory $T^{(i)}$ into a sequence of skill segments, visible in Figures 2b and 2c.

### 4.2. Symbolic Abstraction and Grammar Induction

Once segmented, we abstract the data to identify higher-level composition. This involves two steps: corpus construction and hierarchy induction.

We collapse contiguous frames (steps) sharing the same label into single atomic symbols. A trajectory $z_{1:n}^{(i)}$ is re-

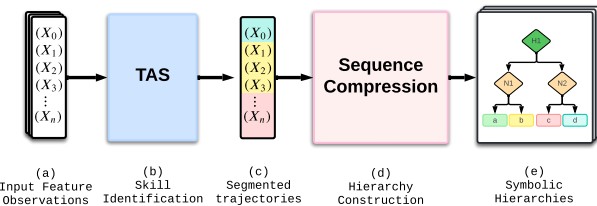

*Figure 2.* Overview of the HiSD pipeline. Demonstration trajectories are first segmented into skills. These skill sequences are then compressed and structured using a modified Sequitur algorithm, which identifies recurring mid-level subroutines across the dataset. The resulting grammar defines a hierarchical task decomposition.

duced to a sequence $S^{(i)} = (\alpha_1, \alpha_2, \ldots, \alpha_{m_i})$, where each $\alpha_j \in \Sigma$ is a terminal symbol corresponding to one of the $K$ discovered skills. This removes the variance of duration, allowing the model to focus purely on the structural sequencing of skills.

Standard grammar induction operates on a single sequence. To learn subroutines that generalise across episodes, we construct a unified corpus $\mathcal{S}_{corp}$ by concatenating all episode sequences separated by a unique boundary token $\phi$: $\mathcal{S}_{corp} = S^{(1)} \oplus \phi \oplus S^{(2)} \oplus \cdots \oplus \phi \oplus S^{(N)}$. We run the Sequitur algorithm on $\mathcal{S}_{corp}$. We explicitly modify the induction constraints to forbid the boundary token $\phi$ from being included in any production rule. This ensures that learned non-terminals capture behaviours internal to episodes, preventing the grammar from merging the end of one episode with the start of another. This is shown in Figure 2d.

The result is a global grammar $G$ where terminal nodes correspond to the atomic skills found in Stage 1 (Figure 2c), and internal nodes represent discovered subroutines (such as "Collect Wood" + "Make Workbench"). By parsing an episode $S^{(i)}$ using $G$, we generate a hierarchical tree $\tau^{(i)}$ that decomposes the task into its constituent sub-goals. This final hierarchy is evident in Figure 2e.

# 5. Experiments and Domains

Our experiments evaluate the ability of HiSD to discover reusable skills and compositional structure in both fully observable (Craftax) and partially observable (Minecraft) domains. These environments present long-horizon challenges well-suited for analysing hierarchical reuse. To ensure baselines are evaluated under their most favourable conditions, we provide CompILE and OMPN with ground truth sub-task orderings, and we standardise the setting by providing the maximum number of skills, $K$, to all models. Beyond this, the methods differ substantially in the supervision they require: CompILE additionally requires action labels and the number of segments per trajectory, OMPN requires action labels and the depth of the hierarchy, while

HiSD operates without these supervisory signals (requiring only $K$). Table 1 summarises this asymmetry.

| Method | Actions | Sub-task Order | $K$ | Other |
|---|---|---|---|---|
| **HiSD (Ours)** | — | — | ✓ | — |
| CompILE | ✓ | ✓ | ✓ | segments per trajectory |
| OMPN | ✓ | ✓ | ✓ | hierarchy depth |

*Table 1.* Supervision required by each method during skill/hierarchy discovery. All methods receive the maximum number of skills $K$. HiSD requires no further supervision, whereas baselines additionally require action labels, ground truth sub-task orderings, and structural priors (segment counts or hierarchy depth).

In this work, we refer to "ground truth" labels not as universal truths, but as algorithmic annotations derived from domain knowledge. For both environments, we construct these labels programmatically by monitoring game state changes (specifically inventory deltas and interaction logs) to assign skill labels to trajectory segments. For Craftax and Minecraft (Matthews et al., 2024; Baker et al., 2022), the approximate number of distinct skills can be inferred from the game's compositional structure: each distinct event, such as gathering a resource, crafting an item, or placing a block, corresponds to an individual skill.

## 5.1. Craftax Environment

Craftax (Matthews et al., 2024) is a 2D top-down environment inspired by Minecraft, where agents gather resources, craft tools, and complete long-horizon survival-themed tasks in a procedurally generated world. We modify the environment to make it fully observable and deterministic, yielding a Markovian environment where the agent can observe the full state at every timestep. Our modified Craftax environment provides a rendered $274 \times 274 \times 3$ RGB image of the top-down view of the game. The action space is reduced from the original game, and consists of 4 cardinal movements and 12 non-movement actions covering interaction, placement, and crafting. We generate a variety of tasks in the Craftax domain, each described in detail below. For each task there is a "static" and a "random" configuration: "static" configurations are those where skills always occur in the same order, while "random" configurations have variation in the skill sequencing between episodes. Each task instance includes a randomly generated world layout and a hand-crafted goal. For a visual example and full discussion of the modified environment, refer to Appendix A.1. We generate 500 expert trajectories per task configuration using an $A^*$ planner guided by domain-specific heuristics. This setting allows us to evaluate whether HiSD can extract consistent skills and meaningful subroutines across diverse episodic expert demonstrations. We implement the following tasks:

- **Stone Pickaxe (Static and Random)**: Requires collecting wood, building a workbench, making a wooden pickaxe, then collecting stone to craft a stone pickaxe. We evaluate both fixed and stochastic sub-task orderings. There are 5 unique skills here.

- **Wood-Stone Collection (Random)**: Involves gathering wood and stone without tool requirements. In this configuration, we collect wood and stone, twice each, in any order. This consists of 2 unique skills.

- **Mixed Task (Static)**: A set of 6 goal types, each requiring a different sequence of skills such as collecting wood, crafting a wooden pickaxe, gathering stone, or building a stone sword. In total, the dataset has 5 unique skills with varying orderings across trajectories.

To obtain compact feature representations from the raw pixel observations, we employ principal component analysis (PCA) (Pearson, 1901; Hotelling, 1933). We fit a separate PCA model for each task using the aggregated dataset of all observation frames, retaining 650 components to capture approximately 99% of the total variance.

### 5.2. Minecraft Environment

To evaluate HiSD in a highly realistic and challenging setting, we turn to the full, unmodified version of Minecraft (Guss et al., 2019; Baker et al., 2022), which presents a partially observable, long-horizon environment with complex low-level controls. Unlike prior work that uses simplified versions of Minecraft where crafting is treated as a discrete action (Kanervisto et al., 2020), we use the native game interface without changes. This requires the agent to interact via raw keyboard and mouse inputs. The agents operate using only raw pixel observations of size $640 \times 360$ and issue low-level control actions. Rather than relying on human expert data, we collect successful trajectories using OpenAI's pre-trained VPT (Video PreTraining) models (Baker et al., 2022). We generate 500 episodes where the agent collects two stone blocks. Unlike the A* planner in Craftax, the VPT policy is not optimal; the resulting demonstrations are noisy and exhibit human-like sub-optimality, including redundant actions and wandering. This provides a test of HiSD's ability to extract structure from imperfect data in partially observable domains.

We construct our dataset by halting the agent once it has collected two blocks of stone. Ground truth annotations are extracted algorithmically by cross-referencing inventory deltas with interaction logs (such as block interaction and destruction events) to map frames to specific skills. Visual features are extracted from first-person observations using MineCLIP (Fan et al., 2022), which produces 512-dimensional CLIP-like embeddings. From this data, we curate two skill-labelled datasets: (1) an *All* dataset containing 44 skills, and (2) a *Mapped* dataset with 14 high-level categories grouping semantically similar skills. For a visual example, refer to Appendix A; for implementation details regarding the skills and ground truth, refer to Appendix A.2.

### 5.3. Skill Metrics

We evaluate segmentation using three standard TAS metrics: **Mean-over-Frames (MoF)**, measuring frame-wise accuracy but sensitive to class imbalance; **F1 Score**, a segment-level metric (overlap $> 50\%$) less biased toward frequent classes; and **mean Intersection-over-Union (mIoU)**, which robustly handles imbalance by strictly penalising over- and under-segmentation. We compute these under two matching schemes: *Per* (local alignment per episode) and *Full* (global Hungarian alignment). We prioritise *Full* mIoU to assess globally reusable skills, contrasting with baselines like CompILE and OMPN that report strict boundary-based F1 scores ($\pm 1$ timestep) focused only on local segmentation quality. We adopt standard TAS (IoU-based) metrics as they are more robust to temporal noise and provide deeper insight into the semantic, cross-episode consistency of the discovered skills (Xu & Gould, 2024; Lea et al., 2017; Lu et al., 2021; Kipf et al., 2019).

### 5.4. Hierarchy Metrics

To assess structural quality and reuse, we compute tree-level metrics averaged over the dataset and compare them relative to the ground truth structure (generated by running our modified Sequitur on clean ground truth labels). We report: (1) **Unique Trees**, where a lower count indicates consistent, reusable decomposition across episodes; (2) **Average Depth**, measuring the level of temporal abstraction to detect under- or over-decomposition; (3) **Average Size** (total nodes), evaluating representational parsimony relative to the task's logical complexity; and (4) **Branching Factors** (mean/max children per node), reflecting the granularity of the decomposition, where high branching suggests a lack of intermediate subroutines.

## 6. Results and Discussion

This section presents our experimental evaluation across both the Craftax and Minecraft domains. We analyse performance in two stages: first, we assess the *skill segmentation* capabilities of each framework using standard quantitative metrics, supplemented by qualitative visualisations. Second, we evaluate the *structural quality* of the hierarchies induced by OMPN and HiSD using tree-level metrics and representative visual outputs.

To ensure a rigorous comparison, we conduct an extensive hyperparameter sweep for all methods; the search ranges

and final selected parameters are detailed in Appendix E. While the quantitative results in Table 2 report averages over 5 random seeds, the qualitative hierarchy analysis utilises the single best-performing run for each framework (selected via mIoU) to illustrate peak representational capacity.

Due to space constraints, we present a curated subset of visualisations in this section; the complete catalogue of discovered skills and hierarchies is available in Appendix D, alongside a computational resource analysis in Appendix B. Finally, we examine the sensitivity of our method to the skill budget, $K$, in Appendix D. Our results indicate that while the performance of HiSD is maximised when $K$ aligns with the ground truth, the framework is robust to mis-specification; particularly in cases of over-estimation, we observe that performance degrades gracefully rather than suffering catastrophic collapse.

### 6.1. Craftax Results

#### 6.1.1. SKILLS

Table 2 presents the skill metrics discussed above. On the simpler WSWS Random task, we see that both baselines perform relatively well in comparison to HiSD. However, as task complexity increases (both in length and number of skills), HiSD outperforms the baselines. In the Stone Pickaxe Static task, which remains relatively simple, both OMPN (Lu et al., 2021) and CompILE (Kipf et al., 2019) struggle to achieve 50% Avg. mIoU. HiSD's advantage is also evident in Figure 3: in this evaluation, both baselines were supplied with the ground truth sub-task order at inference time, yet they still fail to accurately detect skill boundaries; CompILE, in particular, omits some skills entirely. Qualitatively, we observe that HiSD learns to group perceptually distinct but semantically equivalent behaviours into a single skill cluster. For example, in Craftax and Minecraft, visually distinct approaches to the same object (such as collecting wood from different directions or angles) are consistently assigned to the same skill cluster. For a visual example of this, refer to Appendix D.2.

#### 6.1.2. HIERARCHY

Table 3 presents the results for the tree metrics. Ideally, the learned hierarchies should reflect task complexity through appropriate depth, generate consistent structures across similar episodes, and adaptively vary where skill sequencing differs. The number of unique trees should also align closely with the ground truth. This holds for HiSD on the simplest tasks; for example, in WSWS Random it matches the ground truth exactly (9 unique trees). In contrast, OMPN produces a different hierarchy for nearly every episode, showing no structural consistency. Its hierarchies are also larger in size than those from HiSD, despite having similar depth, suggesting excessive branching at each level.

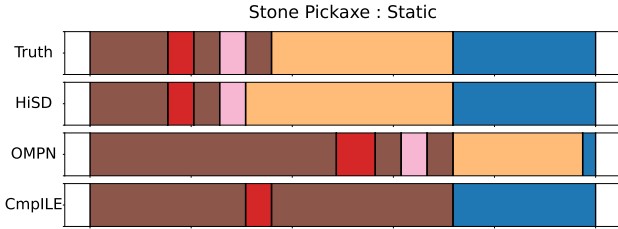

*Figure 3.* Example of the skill segmentation performance in the Stone Pickaxe Static Task in Craftax for all three baselines. Colours indicate discovered skills: wood, table, wooden pickaxe, stone, and stone pickaxe.

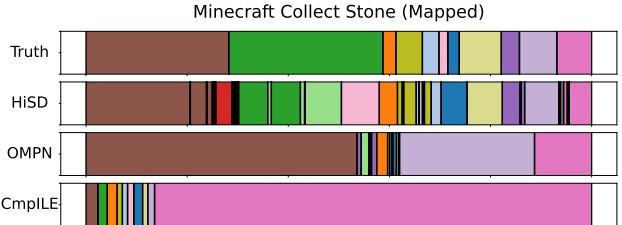

*Figure 4.* Example of the skill segmentation performance in the Minecraft Mapped Task for all three approaches. Colours indicate discovered skills: Walk, Mine Log, Craft Planks, Craft Table, Craft Stick, Use Table, Craft Wooden Pickaxe, Mine Table, Mine Grass, Mine Dirt, and Mine Stone.

### 6.2. Minecraft Results

We now present the same analysis in the Minecraft domain. To enable OMPN and CompILE to operate in this setting, we modify the dataset due to their reliance on discrete action representations during training. Specifically, we discretise the action space by collecting the unique set of all actions observed in the original dataset. This results in 2385 unique integer actions for the domain. By contrast, HiSD does not require any action information and therefore operates directly on observational trajectories.

#### 6.2.1. SKILLS

Referring again to Table 2, HiSD outperforms both baselines. This is a particularly strict test of observation-only segmentation: in Minecraft, walking transitions account for roughly 6% of frames, and walking-toward-wood is visually indistinguishable from walking-toward-stone in first-person view, yet the two correspond to different goals. Despite this perceptual aliasing, HiSD's temporal consistency prior allows it to assign these segments coherently based on surrounding context. In the "All" setting, it achieves significantly higher mIoU, indicating more accurate skill segmentation and identification despite the class imbalance inherent to the Minecraft environment. A qualitative example of the segmentation on the Mapped task is shown in Figure 4.

| Task | Framework | Avg. mIoU | F1 Per | F1 Full | mIoU Per | mIoU Full | MoF Per | MoF Full |
|---|---|---|---|---|---|---|---|---|
| **Craftax** WSWS Random | HiSD | 63% (±12) | 88% (±6) | 74% (±11) | 68% (±9) | 58% (±16) | 79% (±7) | 73% (±14) |
| | OMPN | 75% (±12) | 96% (±8) | 91% (±9) | 78% (±12) | 72% (±11) | 86% (±8) | 83% (±8) |
| | CompILE | **76% (±4)** | **100% (±0)** | **94% (±3)** | **78% (±4)** | **74% (±4)** | **86% (±3)** | **85% (±3)** |
| **Craftax** Stone Pickaxe Static | HiSD | **66% (±15)** | **83% (±12)** | **82% (±13)** | **67% (±13)** | **65% (±17)** | **74% (±7)** | **72% (±11)** |
| | OMPN | 30% (±4) | 65% (±6) | 56% (±4) | 35% (±5) | 26% (±4) | 59% (±2) | 49% (±3) |
| | CompILE | 45% (±18) | 72% (±17) | 67% (±20) | 50% (±17) | 40% (±18) | 70% (±14) | 64% (±16) |
| **Craftax** Stone Pickaxe Random | HiSD | **59% (±2)** | **79% (±2)** | **74% (±4)** | **63% (±1)** | **56% (±3)** | **73% (±1)** | **69% (±3)** |
| | OMPN | 27% (±3) | 57% (±5) | 44% (±1) | 32% (±5) | 21% (±2) | 60% (±2) | 48% (±2) |
| | CompILE | 32% (±7) | 56% (±8) | 52% (±4) | 35% (±10) | 29% (±3) | 65% (±6) | 58% (±3) |
| **Craftax** Mixed Static | HiSD | **62% (±10)** | **81% (±6)** | **47% (±8)** | **71% (±10)** | **53% (±11)** | 74% (±3) | 64% (±5) |
| | OMPN | 49% (±9) | 77% (±4) | 36% (±7) | 64% (±5) | 34% (±13) | 80% (±3) | 63% (±8) |
| | CompILE | 56% (±8) | 81% (±7) | 39% (±6) | 70% (±8) | 41% (±8) | **81% (±5)** | **65% (±7)** |
| **Minecraft** All | HiSD | **31% (±2)** | **55% (±5)** | **18% (±3)** | **49% (±3)** | **12% (±1)** | 38% (±2) | **32% (±3)** |
| | OMPN | 14% (±6) | 29% (±14) | 7% (±3) | 21% (±10) | 7% (±3) | **54% (±10)** | 29% (±6) |
| | CompILE | 6% (±3) | 18% (±6) | 4% (±1) | 10% (±5) | 2% (±1) | 36% (±5) | 19% (±5) |
| **Minecraft** Mapped | HiSD | **38% (±5)** | **64% (±6)** | **53% (±7)** | **43% (±6)** | **33% (±5)** | 51% (±4) | **49% (±4)** |
| | OMPN | 14% (±6) | 28% (±13) | 15% (±6) | 19% (±9) | 8% (±4) | **54% (±10)** | 29% (±5) |
| | CompILE | 6% (±1) | 14% (±2) | 11% (±2) | 6% (±1) | 5% (±2) | 35% (±4) | 34% (±3) |

*Table 2.* Comparison of different skill segmentation approaches. Higher is better for all metrics. Each entry reports the mean performance over five runs, with the corresponding 95% confidence interval (shown in parentheses). "Full" denotes global alignment across all episodes, while "Per" denotes per-episode alignment. When comparing average mIoU, HiSD outperforms all baselines on most tasks, particularly those with added stochasticity or longer horizons. Ties are broken by smaller error.

### 6.2.2. HIERARCHY

Evaluating the hierarchy shows the same trends as Craftax, except now we see much larger trees discovered due to the stochastic and noisy Minecraft environment. Looking at the "All" task that employs all 44 skills, we see that both HiSD and OMPN produce a unique tree per episode. In this case, even the ground truth contains 293 unique trees, showing the amount of noise present in the dataset. This is mirrored by the size of the trees discovered, where the size is inflated by the stochasticity of the environment, leading to far more skills being identified than required. Ultimately, inconsistencies in the initial skill segmentation stage, amplified by the environment's complexity, produce highly variable symbolic sequences that prevent our deterministic grammar from discovering a consistent underlying structure. However, this trend does not necessarily continue in the "Mapped" task where there are fewer skills to reason with. Here, while HiSD still produces a unique tree per episode, the size and the branching factors of the trees are far reduced from the "All" task, which allows us to both visualise and understand the structure of the tree when analysing it qualitatively.

## 7. Reinforcement Learning Deployment

We demonstrate the utility of the discovered hierarchies in downstream RL. While discovery is observation-only, we assume action labels are available during the downstream phase to ground symbolic skills into executable policies via Behavioural Cloning (BC); alternatively, methods like Behavioural Cloning from Observation (Torabi et al., 2018) could infer actions from observations and online interaction. This distinguishes our approach from baselines like OMPN that do not natively support modular option transfer. We formulate discovered skills as options (Sutton et al., 1999). For each skill, we train a low-level policy $\pi_i(a|s)$ using BC and learn initiation $\mathcal{I}_i$ and termination $\beta_i(s)$ conditions via positive-unlabelled (PU) classifiers (Elkan & Noto, 2008) trained on the segmented observations. Intermediate hierarchy nodes function as composite options, executing child nodes sequentially according to the induced grammar. A high-level agent trained with Maskable Proximal Policy Optimization (PPO) (Schulman et al., 2017) outputs a categorical distribution over these valid options at each timestep, using the learned initiation sets as action masks. For full implementation details, refer to Appendix C.

### 7.1. Craftax Evaluation

We evaluate agents on a new "Craft Wooden Pickaxe" task, a sub-goal of the larger "Craft Stone Pickaxe" task involving wood collection and crafting sequences for a sparse reward of +1. We compare PPO agents using HiSD skills and hierarchies against OMPN, CompILE, primitive-action PPO, and pure imitation learning baselines.

| Task | Framework | Unique Trees ↓ | Depth | Size | Avg. Branching | Max Branching |
|------|-----------|----------------|-------|------|----------------|---------------|
| **Craftax** WSWS Random | Truth | 9 | 3.98 | 5.96 | 1.64 | 2.00 |
|  | HiSD | 9 | 3.21 | 4.70 | 1.54 | 2.00 |
|  | OMPN | 499 | 3.34 | 6.73 | 2.04 | 2.47 |
| **Craftax** Stone Pickaxe Static | Truth | 1 | 3.00 | 9.00 | 4.00 | 7.00 |
|  | HiSD | 36 | 6.02 | 13.44 | 1.85 | 2.00 |
|  | OMPN | 500 | 4.94 | 16.20 | 2.05 | 4.54 |
| **Craftax** Stone Pickaxe Random | Truth | 7 | 5.19 | 11.93 | 1.97 | 2.84 |
|  | HiSD | 47 | 5.49 | 12.84 | 1.86 | 2.03 |
|  | OMPN | 500 | 4.88 | 15.54 | 2.01 | 5.10 |
| **Craftax** Mixed Static | Truth | 5 | 3.53 | 5.66 | 1.57 | 2.07 |
|  | HiSD | 13 | 4.31 | 8.14 | 1.59 | 1.81 |
|  | OMPN | 391 | 3.11 | 7.53 | 1.92 | 2.84 |
| **Minecraft** All | Truth | 293 | 8.15 | 22.69 | 1.97 | 2.25 |
|  | HiSD | 500 | 8.48 | 283.25 | 2.21 | 28.73 |
|  | OMPN | 500 | 4.90 | 113.19 | 7.88 | 43.87 |
| **Minecraft** Mapped | Truth | 151 | 8.15 | 21.36 | 2.03 | 3.01 |
|  | HiSD | 500 | 8.52 | 83.85 | 2.08 | 5.40 |
|  | OMPN | 500 | 4.90 | 113.19 | 7.88 | 43.87 |

*Table 3.* A table showing the comparison between trees discovered by: ground truth, HiSD and OMPN. We see in general that HiSD discovers a more consistent decomposition, while generally resulting in smaller, more interpretable trees.

Figure 5 shows mean episode rewards across ten random seeds. The HiSD Hierarchy agent demonstrates superior sample efficiency, reaching near-optimal rewards within ∼30k steps. Crucially, it outperforms the HiSD Skills-Only agent, confirming that compositional structure provides a necessary inductive bias for temporal credit assignment.

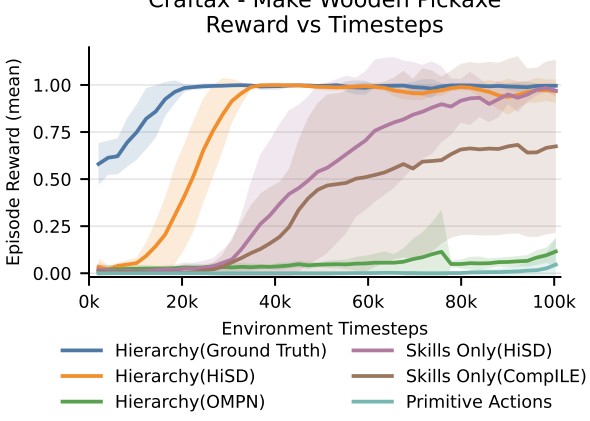

*Figure 5.* Mean episode rewards (±1 SD) over environment timesteps for 10 random seeds on the Craftax Wooden Pickaxe task. The HiSD hierarchy (orange) achieves higher and more stable performance than the OMPN hierarchy (green) and the Skills-Only variants for both HiSD (pink) and CompILE (brown), while closely matching the ground truth hierarchy (blue). Primitive action PPO (cyan) fails to solve the task.

In contrast, the OMPN hierarchy struggles to converge, likely due to excessive branching factors inflating the action space, and PPO using only primitive actions fails to solve the task entirely. CompILE Skills-Only similarly underperforms the HiSD hierarchy, reaching roughly 0.7 reward. Standard imitation learning (not shown) also fails, achieving 0 reward. We further investigate how many action-labelled demonstrations are needed to deploy the discovered structures: a sweep over $N \in \{10, 20, \ldots, 500\}$ episodes (Appendix C.3) shows that the HiSD hierarchy reaches $0.94 \pm 0.08$ mean reward by $N = 250$ and matches the GT hierarchy from $N = 350$ onward, with the segmentation stage itself remaining fully unsupervised throughout. This indicates that although policy grounding requires action labels, the discovered hierarchy works in a low-data regime, with relatively few demonstrations sufficing to recover near-optimal performance.

### 7.2. Minecraft Evaluation

We further validate our approach in Minecraft on a "Collect Log" task using the Mapped dataset, comparing the HiSD Hierarchy against HiSD Skills-Only, primitive action, and ground truth baselines. Results in Minecraft (Figure 6) mirror Craftax. Primitive action PPO fails due to the horizon length. However, the HiSD Hierarchical agent successfully solves the task 50% of the time. While Ground Truth yields the highest performance, HiSD's hierarchy outperforms the flat HiSD Skills-Only agent, confirming that hierarchical structure enables RL in complex environments.

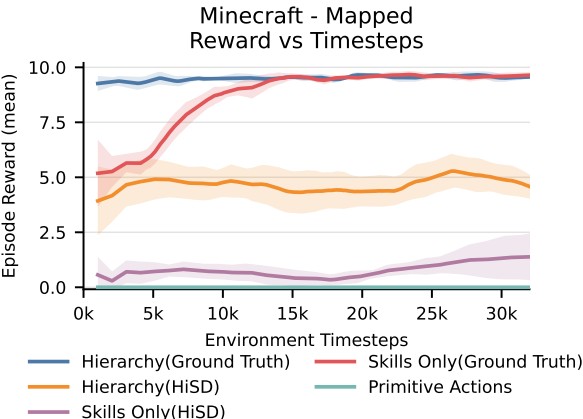

*Figure 6.* Mean episode rewards (±1 SD) over environment timesteps for 5 random seeds on the Minecraft Collect Log task. The HiSD hierarchy (orange) achieves higher performance than the Skills-Only (HiSD) variant (purple). We note that both fall short of the ground truth setups that manage to solve the task 100% of the time. Primitive action PPO (cyan) again fails to solve the task.

## 8. Limitations and Future Work

HiSD operates on pre-extracted feature representations and assumes a prior on the maximum number of skills $K$; integrating representation learning end-to-end is a natural extension. While we treat $K$ as a fixed hyperparameter, it could also be inferred adaptively by running HiSD with a decreasing schedule of $K$ from a high initial estimate, using indicators such as the skill-label switching frequency, the number of active clusters at convergence, or the aggregate segmentation cost as a stopping criterion. A further structural assumption is that each timestep is assigned to a single discrete skill: this is standard in temporal action segmentation and is shared by our baselines, but settings with naturally concurrent behaviours (e.g., robotic manipulation where grasping and locomotion overlap) may benefit from multi-label or factored skill representations (e.g., via multi-label optimal transport), which we leave to future work. A more fundamental limitation is that Sequitur is deterministic and cannot absorb segmentation noise. As a result, near-duplicate skill sequences are treated as distinct, preventing reuse of subroutines, as seen in Minecraft "All" where HiSD produces 500 unique trees for a shared task. Probabilistic approaches such as PCFGs (Lari & Young, 1990) or fragment grammars (O'Donnell et al., 2009), potentially augmented with learned action effects, could address this by marginalising over noisy parses and capturing causal structure.

## 9. Conclusion

We introduce HiSD, a unified framework for unsupervised skill segmentation and hierarchical structure discovery in re-inforcement learning domains. By bridging state-of-the-art temporal action segmentation with grammar-based compression, HiSD induces multi-level hierarchies directly from raw observational trajectories. Unlike prior methods that rely on action supervision, reward signals, or manual segmentation (Ranchod et al., 2015; Lu et al., 2021; Kipf et al., 2019), our approach operates without these constraints, requiring only a loose prior on the maximum number of skills. Evaluated on the complex domains of Craftax and Minecraft, HiSD consistently outperforms baselines such as CompILE and OMPN, producing hierarchies that are deeper, more reusable, and semantically interpretable. Crucially, we show that meaningful structure can be recovered purely from temporal coherence and visual similarity. As a proof of concept for utility, we demonstrate that when these discovered abstractions are used in downstream RL, they significantly accelerate learning, achieving performance comparable to ground truth hierarchies while improving stability and enabling RL in complex, high-dimensional environments. Our results suggest that unsupervised structure discovery offers a scalable path toward generalist agents capable of learning from abundant, unlabelled data.

## Acknowledgements

Computations were performed using the High Performance Computing (HPC) infrastructure provided by the Mathematical Sciences Support unit at the University of the Witwatersrand, Johannesburg. This work was supported in part by the National Research Foundation (NRF) of South Africa through the Thuthuka Funding Instrument (Grant Number: TTK240416214385).

## Impact Statement

This work contributes to the fields of reinforcement learning and imitation learning by proposing a method for discovering skill hierarchies from observational data. Our approach lowers the barrier to training capable agents by removing the need for dense supervision and action labels. The generated hierarchies offer a degree of transparency often lacking in RL, facilitating better human-AI interaction. We do not foresee any immediate negative societal consequences resulting directly from this fundamental research, though standard ethical considerations regarding the deployment of autonomous decision-making systems apply.

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

# Appendix

This section outlines and provides the supplementary information and figures for the main paper on *Unsupervised Hierarchical Skill Discovery*. We note that all code written and used is available via our `GitHub Repository`.

## A. Environment Details and Examples

Figure 7 provides examples of the observations from both Craftax (Matthews et al., 2024) and Minecraft (Baker et al., 2022) environments.

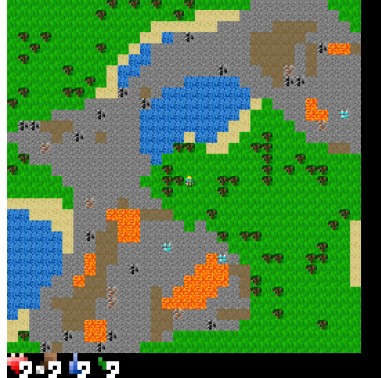

**(A)** Example of an RGB observation from the modified Craftax environment

**(B)** Example of an RGB observation from the Minecraft environment

*Figure 7.* Environment observation examples from Craftax (Matthews et al., 2024) (A) and Minecraft (Baker et al., 2022) (B).

### A.1. Craftax

We model the Craftax environment (Matthews et al., 2024) as a fully observable Markov Decision Process (MDP), diverging from its standard Partially Observable Markov Decision Process (POMDP) formulation. To achieve this, we introduce specific modifications that simplify the dynamics and grant the agent global visibility:

1. **Global Observation:** The camera is fixed to capture the entire world state, removing partial observability.

2. **Stationary Mechanics:** Day/night cycles are disabled to maintain consistent lighting.

3. **Simplified Survival:** Health, energy, and food decay are disabled, giving the agent infinite stamina.

4. **Static Environment:** All autonomous entities (monsters and animals) are removed.

Each observation returned by the environment is a $274 \times 274 \times 3$ RGB image representing a top-down view of the full $32 \times 32$ map (see Figure 7A). The action space is restricted to 16 discrete primitives, categorised by functionality in Table 4.

| Category | IDs | Actions |
|---|---|---|
| Movement | 0, 1-4, 6 | No-op, Move (L/R/U/D), Sleep |
| Interaction | 5 | Do Action (Break Block) |
| Placement | 7-10 | Place : Stone, Workbench, Furnace, Plant |
| Crafting | 11-15 | Craft : Pickaxes (Wood/Stone/Iron), Swords (Stone/Iron) |

*Table 4.* The condensed action space used in our modified Craftax environment.

### A.1.1. CRAFTAX TASK AND SKILL STATISTICS

This section analyses the expert demonstrations collected for the Craftax environment. Figure 8 illustrates the distribution of skills employed across varying tasks. This metric is derived by summing the number of steps (frames) a specific skill is active over the entire dataset. We observe that all datasets (with the exception of the 2-skill WSWS Random dataset) exhibit a long-tail distribution. In these cases, the "wood" interaction skill dominates, reflecting its role as a fundamental, high-frequency discriminant action within the environment.

Additionally, we analyse task complexity and prerequisites. Table 5 details the episode length statistics. As expected, complex tasks correlate with increased episode duration; simpler tasks average approximately $14 - 16$ steps, whereas harder tasks average roughly 33 steps. Finally, Table 6 outlines the specific item dependencies and resource requirements necessary to successfully complete each task.

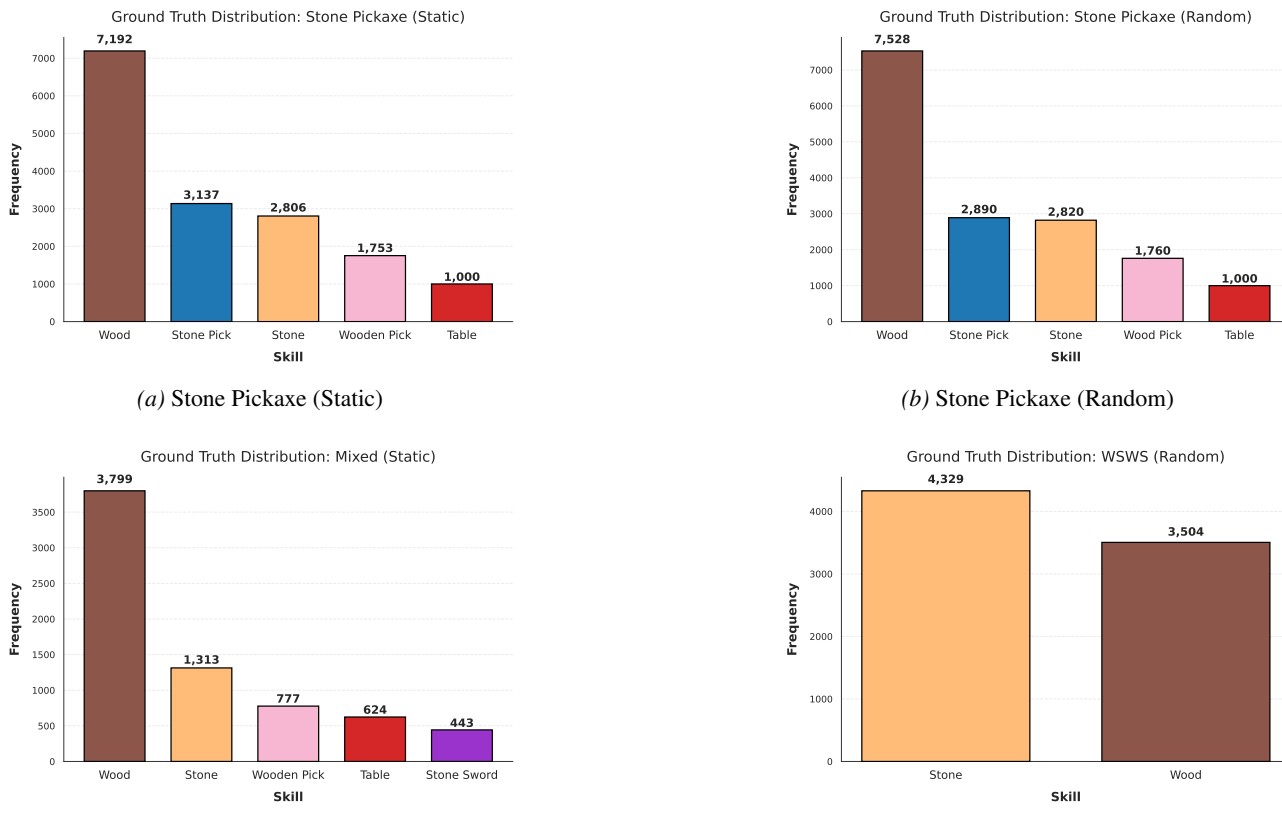

*(a)* Stone Pickaxe (Static)

*(b)* Stone Pickaxe (Random)

*(c)* Mixed Task (Static)

*(d)* Wood-Stone Collection (Random)

*Figure 8.* **Ground truth skill distributions.** The histograms depict the frequency of skill usage across different Craftax tasks. The 'wood' skill serves as a primary interaction mechanic in most configurations.

| Task Name | Min | Avg | Max |
|---|---|---|---|
| Stone Pickaxe (Static) | 17 | 32.78 | 102 |
| Stone Pickaxe (Random) | 19 | 33.00 | 74 |
| Mixed Task (Static) | 3 | 14.91 | 58 |
| Wood-Stone Coll. (Rand) | 10 | 16.67 | 42 |

*Table 5.* Episode length statistics for expert demonstrations.

| Target | Ingredients |
|---|---|
| Wood | Break $1\times$ Tree |
| Workbench | $2\times$ Wood |
| Wood Pick | $1\times$ Wood + Bench |
| Stone | Mine w/ Wood Pick |
| Stone Pickaxe | $1\times$ Wood + $1\times$ Stone |
| Stone Sword | $1\times$ Wood + $1\times$ Stone |

*Table 6.* Crafting dependencies.

## A.2. Minecraft

We adopt the Minecraft environment specification from MineRL (Guss et al., 2019), modelled as a POMDP. The agent operates on high-dimensional sensory input, receiving RGB frames of size $640 \times 360 \times 3$ at each timestep. To extract semantic features for our experiments, these raw visual observations are encoded using the MineCLIP (CLIP4MC) video-text encoder (Fan et al., 2022).

The agent utilises the standard MineRL action space, which corresponds to low-level keyboard and mouse commands. This includes navigation, interaction, and GUI management (required for crafting items). The action space is a dictionary of binary triggers (such as movement, or hotbar selection) and discretised camera controls. A summary of these action groups is provided in Table 7. The camera bins used are: $\{0°, \pm 0.62°, \pm 1.61°, \pm 3.22°, \pm 5.81°, \pm 10°\}$

| Category | Actions Included | Type |
|---|---|---|
| Movement | Forward, Back, Left, Right, Jump, Sprint, Sneak | Binary |
| Interaction | Attack, Use, Drop, Pick Item, Swap Hands | Binary |
| Interface | Inventory, Escape (Menu/ No-Op) | Binary |
| Hotbar | Slots 1–9 | Binary |
| Camera | Pitch, Yaw | Discrete (11 bins each) |

*Table 7.* Summary of the standard MineRL action space. The camera actions are discretised into 11 distinct bins (Guss et al., 2019).

### A.2.1. MINECRAFT TASK AND SKILL STATISTICS

This section analyses the expert demonstrations collected for the Minecraft environment. Figure 9 illustrates the distribution of skills employed across the two task configurations: "All" (fine-grained) and "Mapped" (semantically grouped). This metric is derived by summing the number of steps (frames) a specific skill is active over the entire dataset. Consistent with the Craftax analysis in Appendix A.1.1, we observe a long-tail distribution in skill usage.

We further analyse task complexity and prerequisites in Table 8 and Table 9. The left panel details episode length statistics; note that these are identical for both configurations as they share the same underlying trajectory data, differing only in ground truth labelling. The right panel outlines the specific item dependencies and resource requirements necessary for gathering resources and crafting. Note that any `*stone*` or `*ore*` block requires a `Wooden Pickaxe` to be broken and collected; other blocks listed can be mined by hand.

Finally, Table 10 presents the semantic mapping schema used to condense the "All" task space into the "Mapped" space. This aggregation groups visually or functionally similar items (such as all wood log variants) into single skill categories to facilitate segmentation. Items not listed in this table retain their original fine-grained labels.

| Task Name | Min | Avg | Max |
|---|---|---|---|
| Minecraft (Mapped) | 630 | 785.72 | 1078 |
| Minecraft (All) | 630 | 785.72 | 1078 |

*Table 8.* Episode length statistics (steps).

| Target | Ingredients |
|---|---|
| $4\times$ Planks | $1\times$ Log |
| Crafting Table | $4\times$ Planks |
| $4\times$ Sticks | $1\times$ Plank |
| Wood Pick | $3\times$ Plank + $2\times$ Stick |
| Cobble/Stone | Mine w/ Wood Pick |

*Table 9.* Crafting dependencies.

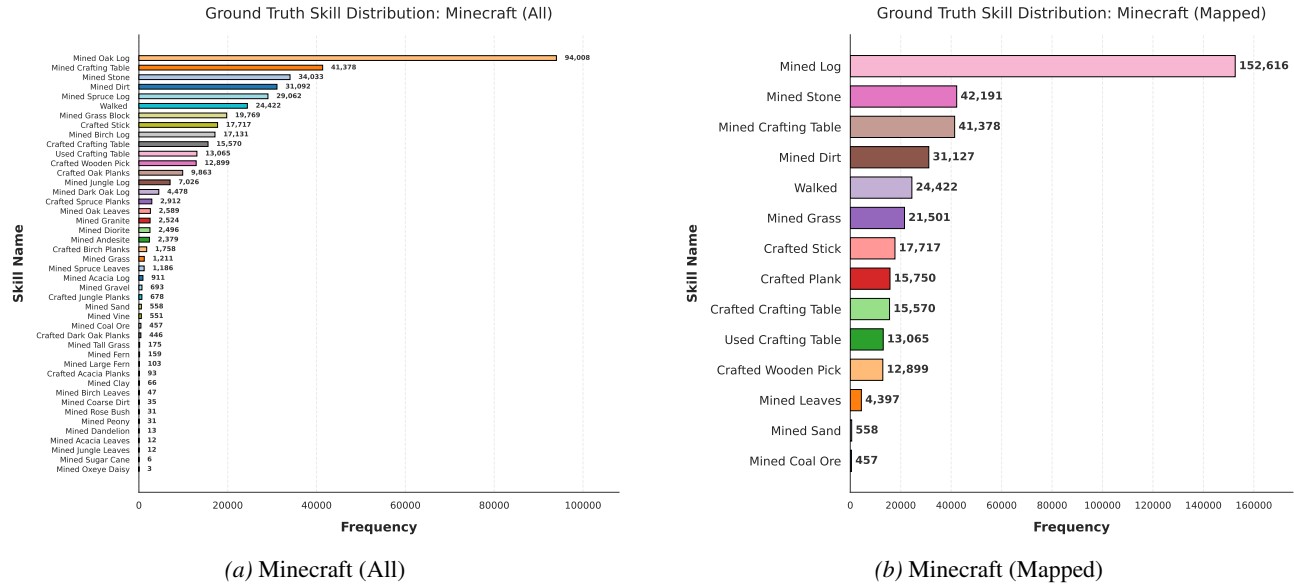

*(a)* Minecraft (All)         *(b)* Minecraft (Mapped)

*Figure 9.* **Ground truth skill distributions for Minecraft Tasks.** The histograms show the frequency of skill usage. Again we see that the Mine Wood skills serve as a primary interaction mechanic in both settings.

| Mapped Category | Fine-Grained Skills (Original Labels) |
|---|---|
| `crafted_plank` | `crafted_acacia_planks`, `crafted_birch_planks`, `crafted_dark_oak_planks`, `crafted_jungle_planks`, `crafted_oak_planks`, `crafted_spruce_planks` |
| `mined_log` | `mined_acacia_log`, `mined_birch_log`, `mined_dark_oak_log`, `mined_jungle_log`, `mined_oak_log`, `mined_spruce_log`, `used_oak_log`, `used_spruce_log` |
| `mined_leaves` | `mined_acacia_leaves`, `mined_birch_leaves`, `mined_jungle_leaves`, `mined_oak_leaves`, `mined_spruce_leaves`, `mined_vines`, `mined_vine` |
| `mined_stone` | `mined_andesite`, `mined_clay`, `mined_diorite`, `mined_granite`, `mined_gravel` |
| `mined_dirt` | `mined_coarse_dirt` |
| `mined_grass` | `mined_large_fern`, `mined_dandelion`, `mined_fern`, `mined_grass`, `mined_grass_block`, `mined_oxeye_daisy`, `mined_peony`, `mined_rose_bush`, `mined_sugar_cane`, `mined_tall_grass` |

*Table 10.* Mapping of fine-grained skills to high-level categories. Skills not listed here (such as `mined_coal`) are retained in their original form.

## B. Hardware Infrastructure and Computational Efficiency

To assess the practical deployability of our method, we contrast the computational resources required for the baselines (CompILE, OMPN) against our proposed method (HiSD).

BASELINE REQUIREMENTS
The baseline models (CompILE (Kipf et al., 2019), OMPN (Lu et al., 2021)) imposed significant computational overhead. Training required a high-end workstation equipped with an **Intel i9-10940X CPU**, **128GB of RAM**, and an **NVIDIA RTX 3090 (24GB VRAM)**.
Despite this powerful hardware, the memory-intensive architectures of OMPN and CompILE necessitated relatively small batch sizes. Consequently, these models exhibited slow convergence and long training times, highlighting scaling limitations.

HiSD EFFICIENCY (OURS)
HiSD was trained on a more modest setup: an **AMD Ryzen 9800X3D**, **32GB of RAM**, and an **NVIDIA RTX 3080 (10GB VRAM)**.
Notably, HiSD is capable of training and inference on consumer-grade GPUs with as little as **6GB VRAM** (RTX 3050). This low memory footprint and faster convergence rate make HiSD a significantly more scalable solution, enabling deployment on standard consumer hardware without the need for enterprise-grade compute clusters.

|  | Baselines (CompILE, OMPN) | Ours (HiSD) |
| --- | --- | --- |
| **GPU VRAM Required** | High ($\sim$24GB) | Low (6–10GB) |
| **System RAM** | 128GB | 32GB |
| **Batch Size Limit** | Restricted (Memory Bound) | Flexible |
| **Training Duration** | Slow | Fast |
| **Min. GPU Spec** | RTX 3090 (or equivalent) | RTX 3050 (Consumer Grade) |

*Table 11.* Comparison of hardware specifications and computational constraints. HiSD achieves comparable or superior performance with significantly lower memory requirements.

## C. Downstream RL Configuration

To evaluate the utility of the discovered structures, we instantiate the discovered skills as temporal options, denoted as $\omega = \langle \mathcal{I}_\omega, \pi_\omega, \beta_\omega \rangle$. We employ a two-stage process: first, we learn the initiation sets ($\mathcal{I}_\omega$) and termination conditions ($\beta_\omega$) using positive-unlabelled learning; second, we learn the intra-option policies ($\pi_\omega$) via Behavioural Cloning.

Hierarchical nodes (non-terminals in the induced grammar) are instantiated as composite options. These nodes do not require separate policy learning; instead, they execute their constituent child options sequentially. A composite option inherits the initiation condition of its first child and the termination condition of its last child, preserving the temporal logic of the discovered hierarchy. However, we note that OMPN does not learn skills for the leaf nodes and instead represents leaf nodes as sequences of primitive actions. Thus, we do not learn a BC model or a PU model for the OMPN hierarchies, only for the segmented skills OMPN produces.

For the PU models, in both Craftax and Minecraft, we report the following metrics: Micro F1-Score, Macro Recall, Macro Precision, and Overall Accuracy. In multi-label environments with significant class imbalance, reporting a single metric is insufficient to capture model efficacy. We prioritise Micro F1-Score to provide an aggregate measure of global system reliability by weighting each instance equally, while Macro Precision and Macro Recall ensure that performance is not driven solely by majority classes, reflecting the model's ability to generalise to rare skills. Finally, Overall Accuracy serves as a baseline for the total reduction in classification error across the entire feature space. We note all PU and BC models were trained with the computational requirements outlined in Table 11 (left) for the baselines.

### C.1. Craftax Skill Configuration

For the Craftax domain, we utilise a combination of feature-based and pixel-based learning. While the structure discovery and option applicability (PU models) operate on lower-dimensional PCA features for efficiency, the control policies (BC and PPO) operate on pixel-based observations to retain spatial precision.

### C.1.1. INITIATION AND TERMINATION (PU LEARNING)

We treat the identification of valid start and end states as a binary classification problem trained on positive and unlabelled data. We employ the Elkan-Noto PU learning algorithm.

- **Input Data:** PCA features reduced from the raw pixel observations ($D = 650$).

- **Classifier:** We use a Support Vector Machine (SVM) as the base estimator with an RBF kernel, a regularisation parameter $C = 10$, and $\gamma = $ scale.

- **Training:** Models are trained using a hold-out ratio of $0.2$. We perform grouped cross-validation (5 folds) to select decision thresholds that maximise the F1 score for each skill.

The performance of the learned initiation and termination models for the ground truth skills and discovered skills are presented in Tables 12a, 12b, 12c, and 12d.

| Metric | PU-Start | PU-End |
|---|---|---|
| Micro F1-Score | 0.867 | 0.570 |
| Macro Recall | 0.926 | 0.780 |
| Macro Precision | 0.785 | 0.543 |
| Overall Accuracy | 0.955 | 0.954 |

*(a)* Ground Truth Skills

| Metric | PU-Start | PU-End |
|---|---|---|
| Micro F1-Score | 0.883 | 0.533 |
| Macro Recall | 0.925 | 0.728 |
| Macro Precision | 0.795 | 0.493 |
| Overall Accuracy | 0.961 | 0.951 |

*(b)* HiSD Discovered Skills

| Metric | PU-Start | PU-End |
|---|---|---|
| Micro F1-Score | 0.519 | 0.142 |
| Macro Recall | 0.733 | 0.592 |
| Macro Precision | 0.342 | 0.195 |
| Overall Accuracy | 0.764 | 0.691 |

*(c)* OMPN Discovered Skills

| Metric | PU-Start | PU-End |
|---|---|---|
| Micro F1-Score | 0.752 | 0.359 |
| Macro Recall | 0.704 | 0.519 |
| Macro Precision | 0.518 | 0.360 |
| Overall Accuracy | 0.906 | 0.935 |

*(d)* CompILE Discovered Skills

*Table 12.* Comparison of PU Start and End Learning results across the different methods.

### C.1.2. INTRA-OPTION POLICIES (BEHAVIOURAL CLONING)

We learn a parametrised policy $\pi_\theta(a|s)$ for each atomic skill using supervised learning on the expert demonstrations.

- **Architecture:** We employ a ResNet-34 backbone pre-trained on ImageNet. The final classification layer is replaced to output logits for the 16 primitive Craftax actions.

- **Input:** Raw top-down RGB observations, resized to $256 \times 256$ and normalised using ImageNet statistics.

- **Optimisation:** We optimise the Cross-Entropy loss using the AdamW optimiser with a learning rate of $3 \times 10^{-4}$ and weight decay of $3 \times 10^{-4}$ over 150 epochs.

### C.1.3. HIGH-LEVEL PPO CONTROLLER

The downstream agent is trained using PPO with a standard Convolutional Neural Network policy.

- **Input:** Raw $64 \times 64$ top-down pixel observations (unlike the BC policies, the PPO agent does not use the ResNet backbone).

- **Action Space:** The action space is discrete and consists of the primitive actions plus the executable options.

- **Masking:** We use Maskable PPO. At each timestep, the validity of a skill option is determined by querying the corresponding PU initiation model on the current observation's PCA features.

## C.2. Minecraft Configuration

In the Minecraft domain, we leverage pre-trained MineCLIP embeddings for all stages of the pipeline to handle the high-dimensional visual complexity. We enforce a global frameskip of 8 across the environment, BC training, and PPO execution.

### C.2.1. INITIATION (PU LEARNING)

Similar to Craftax, we learn initiation sets via PU learning. However, due to the high stochasticity and noise in the Minecraft transitions, we found the termination (end) PU models to be unreliable. Therefore, in this configuration, we rely solely on initiation models to gate skill availability, while termination is handled via fixed time horizons or implicit sub-goal completion.

- **Input Data:** 512-dimensional MineCLIP embeddings.

- **Classifier:** We use a Logistic Regression base estimator with standard scaling, $L2$ regularisation ($C = 10$), and balanced class weights.

- **Training:** We use the Elkan-Noto wrapper with a hold-out ratio of $0.2$. Thresholds are calibrated via stratified group cross-validation to maximise the F1 score.

The performance of the initiation models is reported in Tables 13a and 13b.

| Metric | PU-Start | PU-End |
|---|---|---|
| Micro F1-Score | 0.590 | 0.088 |
| Macro Recall | 0.761 | 0.409 |
| Macro Precision | 0.642 | 0.034 |
| Overall Accuracy | 0.913 | 0.980 |

*(a)* Ground Truth Skills.

| Metric | PU-Start | PU-End |
|---|---|---|
| Micro F1-Score | 0.857 | 0.108 |
| Macro Recall | 0.830 | 0.324 |
| Macro Precision | 0.764 | 0.028 |
| Overall Accuracy | 0.981 | 0.965 |

*(b)* HiSD Discovered Skills

*Table 13.* Comparison of PU Start and End Learning results for Minecraft skills. PU-End results are included to motivate the decision to use initiation models only in Minecraft.

### C.2.2. INTRA-OPTION POLICIES (BEHAVIOURAL CLONING)

To address partial observability and the temporal nature of the MineCLIP features, we utilise recurrent neural networks for the skill policies.

- **Architecture:** A Gated Recurrent Unit (GRU) with a hidden size of 512 and a single layer. The network outputs a multi-discrete action vector (buttons, yaw, and pitch).

- **Input:** Sequences of MineCLIP embeddings with a sequence length of 32 steps.

- **Optimisation:** We use the AdamW optimiser with a learning rate of $3 \times 10^{-4}$. We employ a composite loss function: Binary Cross-Entropy for button presses and Cross-Entropy for discretised camera actions.

### C.2.3. HIGH-LEVEL PPO CONTROLLER

The high-level policy is trained using Maskable PPO with an MLP policy acting directly on the MineCLIP embeddings.

- **Observation Space:** 512-dimensional MineCLIP feature vectors.

- **Action Masking:** Available options are restricted by the PU initiation models.

- **Execution:** Upon selection, a skill option (driven by the GRU BC policy) executes for up to 64 steps (512 wall-clock ticks) or until the episode terminates. Composite options extend this budget proportionally to the number of leaf nodes in their sequence.

## C.3. Sample Efficiency in Action Labels

A natural practical question is whether the discovered structures remain useful when only a small number of action-labelled demonstrations are available for option learning. While skill discovery in HiSD is observation-only, the downstream BC policies do require action labels. To probe this, we conduct a sweep on the Craftax Wooden Pickaxe task in which we vary the number of action-labelled episodes $N$ used to train the BC and PU components, while keeping the downstream PPO budget fixed at 100k environment steps. The sweep covers $N \in \{10, 20, 30, 40, 50, 60, 70, 80, 90, 100, 200, 250, 300, 350, 500\}$ and is run across 10 PPO seeds per configuration. We report the mean episode reward ($\pm 1$ SD) averaged over the final 10% of training steps in Table 14.

| $N$ | GT Skills | HiSD Skills | GT Hierarchy | HiSD Hierarchy |
|---|---|---|---|---|
| 10 | 0.00 ($\pm 0.00$) | **0.20** ($\pm 0.40$) | 0.00 ($\pm 0.00$) | 0.00 ($\pm 0.00$) |
| 20 | **0.10** ($\pm 0.30$) | 0.00 ($\pm 0.00$) | 0.00 ($\pm 0.00$) | 0.00 ($\pm 0.00$) |
| 30 | **0.96** ($\pm 0.04$) | 0.05 ($\pm 0.14$) | 0.50 ($\pm 0.49$) | 0.00 ($\pm 0.01$) |
| 40 | **0.90** ($\pm 0.17$) | 0.11 ($\pm 0.33$) | 0.08 ($\pm 0.22$) | 0.00 ($\pm 0.00$) |
| 50 | 0.00 ($\pm 0.00$) | 0.00 ($\pm 0.00$) | 0.00 ($\pm 0.00$) | 0.00 ($\pm 0.00$) |
| 60 | **0.93** ($\pm 0.12$) | 0.85 ($\pm 0.32$) | 0.77 ($\pm 0.43$) | 0.00 ($\pm 0.00$) |
| 70 | **0.75** ($\pm 0.43$) | 0.00 ($\pm 0.00$) | 0.67 ($\pm 0.46$) | 0.00 ($\pm 0.00$) |
| 80 | 0.87 ($\pm 0.33$) | 0.95 ($\pm 0.11$) | **0.98** ($\pm 0.03$) | 0.86 ($\pm 0.29$) |
| 90 | 0.31 ($\pm 0.37$) | **0.67** ($\pm 0.45$) | 0.18 ($\pm 0.35$) | 0.00 ($\pm 0.00$) |
| 100 | 0.42 ($\pm 0.50$) | **0.82** ($\pm 0.24$) | 0.00 ($\pm 0.00$) | 0.30 ($\pm 0.43$) |
| 200 | 0.56 ($\pm 0.48$) | **0.84** ($\pm 0.31$) | 0.78 ($\pm 0.44$) | 0.11 ($\pm 0.22$) |
| 250 | 0.63 ($\pm 0.48$) | 0.84 ($\pm 0.31$) | **0.95** ($\pm 0.04$) | 0.94 ($\pm 0.08$) |
| 300 | 0.29 ($\pm 0.45$) | 0.74 ($\pm 0.43$) | 0.95 ($\pm 0.04$) | **0.96** ($\pm 0.08$) |
| 350 | 0.70 ($\pm 0.43$) | **0.98** ($\pm 0.02$) | 0.93 ($\pm 0.12$) | 0.98 ($\pm 0.03$) |
| 500 | 0.76 ($\pm 0.40$) | 0.97 ($\pm 0.02$) | **0.99** ($\pm 0.02$) | 0.97 ($\pm 0.05$) |

*Table 14.* Final mean episode reward ($\pm 1$ SD) across 10 PPO seeds on the Craftax Wooden Pickaxe task as a function of the number of action-labelled demonstration episodes $N$ used for BC and PU training. Values are averaged over the final 10% of PPO training steps. The downstream RL budget (100k environment steps) is held fixed across all entries. Bold indicates the highest mean reward in each row.

The results show that the discovered hierarchies remain useful even with far fewer action labels. Both the GT and HiSD hierarchical agents reach near-perfect reward with low variance by $N = 300$ to $500$, and the HiSD Hierarchy is already at $0.94 \pm 0.08$ by $N = 250$, which is a fairly modest budget by current RL standards. The flat-skill variants tend to score above zero at lower $N$ than their hierarchical counterparts. This is expected, since composite options chain BC policies together and small per-skill errors compound across stages. Once the BC stage becomes reliable, the hierarchical configurations pull ahead, matching the trend reported in Section 7.

The intermediate-$N$ regime is clearly non-monotonic. Standard deviations of around $0.4$ to $0.5$ at $N \in \{30, 60, 70, 90, 200\}$ come from bimodal seed-level outcomes: individual PPO runs either solve the task or collapse to zero, with little in between. The full collapse at $N = 50$ across all four configurations suggests this is a property of the BC stage rather than any specific discovery method, since the same pathology hits the ground truth skills and hierarchy. We report the raw numbers without smoothing because we think the bimodality is informative on its own. It indicates that the bottleneck at small $N$ is BC sample complexity, not the quality of the discovered structure. The segmentation stage itself is fully unsupervised and unaffected by $N$.

# D. Evaluation of Discovered Skills and Hierarchies

In this section, we evaluate the semantic quality and structural coherence of the skills learned by HiSD. First, we analyse the method's sensitivity to the skill count parameter $K$ in Appendix D.1. Subsequently, we examine the object-centric nature of the skills discovered during the first phase of training in Appendix D.2. Finally, we present a qualitative comparison of the hierarchies discovered by HiSD against baseline methods, providing visual examples from both the Craftax and Minecraft domains in Appendix D.3.

## D.1. Effects of Varying $K$

We investigate the impact of the skill budget $K$ on segmentation performance across Craftax tasks. As detailed in Table 15, accurately estimating the underlying number of skills is generally required to maximise segmentation quality. In most cases, setting $K$ to the ground truth value yields the best performance across Average mIoU, F1, and MoF metrics.

For example, in the *Stone Pickaxe Static* task, peak performance is achieved at the ground truth of $K = 5$. However, we observe that the method is relatively robust to over-estimation; setting $K = 7$ results in only a marginal decrease in performance metrics. Similarly, for *Stone Pickaxe Random*, performance remains stable across $K \in \{5, 6, 7\}$. This suggests that in stochastic environments with high execution variance, the model can effectively utilise additional skill slots to capture variations of the same underlying behaviour without suffering from collapse.

In contrast, tasks such as *WSWS Random* and *Mixed Static* exhibit higher sensitivity to $K$. We observe a sharper drop in performance when deviating from the optimal value, particularly when $K$ is underestimated. This indicates that for highly structured or repetitive tasks, a precise skill budget is critical to prevent the merging of distinct primitives or the fragmentation of coherent behaviours.

| Task | K | Avg. mIoU | F1 Per | F1 Full | mIoU Per | mIoU Full | MoF Per | MoF Full |
|---|---|---|---|---|---|---|---|---|
| **Craftax** WSWS Random | **2** | 0.72 | 0.84 | 0.93 | 0.71 | 0.73 | 0.83 | 0.83 |
| | 3 | 0.40 | 0.36 | 0.65 | 0.29 | 0.50 | 0.38 | 0.58 |
| | 4 | 0.41 | 0.34 | 0.56 | 0.31 | 0.51 | 0.38 | 0.52 |
| **Craftax** Stone Pickaxe Static | 3 | 0.40 | 0.66 | 0.66 | 0.40 | 0.40 | 0.67 | 0.67 |
| | 4 | 0.40 | 0.66 | 0.66 | 0.40 | 0.40 | 0.67 | 0.67 |
| | **5** | 0.78 | 0.94 | 0.93 | 0.78 | 0.78 | 0.81 | 0.81 |
| | 6 | 0.47 | 0.64 | 0.66 | 0.46 | 0.47 | 0.64 | 0.65 |
| | 7 | 0.76 | 0.82 | 0.83 | 0.76 | 0.76 | 0.71 | 0.72 |
| **Craftax** Stone Pickaxe Random | 3 | 0.33 | 0.55 | 0.60 | 0.30 | 0.36 | 0.60 | 0.66 |
| | 4 | 0.45 | 0.62 | 0.66 | 0.43 | 0.48 | 0.63 | 0.68 |
| | **5** | 0.61 | 0.80 | 0.77 | 0.64 | 0.58 | 0.74 | 0.71 |
| | 6 | 0.58 | 0.66 | 0.77 | 0.52 | 0.65 | 0.61 | 0.70 |
| | 7 | 0.59 | 0.61 | 0.72 | 0.52 | 0.66 | 0.59 | 0.68 |
| **Craftax** Mixed Static | 3 | 0.41 | 0.34 | 0.74 | 0.20 | 0.62 | 0.47 | 0.70 |
| | 4 | 0.51 | 0.39 | 0.77 | 0.35 | 0.68 | 0.57 | 0.72 |
| | **5** | 0.69 | 0.49 | 0.83 | 0.63 | 0.74 | 0.65 | 0.75 |
| | 6 | 0.65 | 0.51 | 0.84 | 0.55 | 0.75 | 0.68 | 0.76 |
| | 7 | 0.49 | 0.36 | 0.72 | 0.40 | 0.57 | 0.59 | 0.69 |

*Table 15.* Ablation study varying the number of skills ($K$) in the HiSD framework. The row corresponding to the ground truth number of skills for each task is highlighted in **bold**. Results are reported for a single random seed.

## D.2. Type of Skills Learnt by HiSD

Figure 10 provides a qualitative visualisation of the skills discovered by HiSD. A key strength of the method is its ability to learn object-centric behaviours that are robust to positional variance. As illustrated, HiSD consistently classifies the interaction with trees as a unified "Wood" skill, regardless of the agent's approach vector or specific location in the grid. Furthermore, the model successfully disentangles semantically distinct actions even in cluttered environments; for instance, it sharply distinguishes the "Table" interaction from "Wood" gathering, even when the workbench is placed in close proximity to the trees.

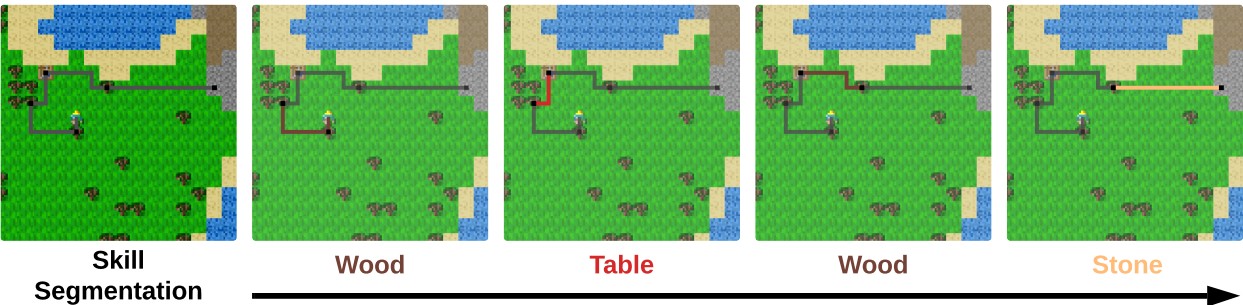

*Figure 10.* **Qualitative visualisation of learned skill primitives in the Stone Pickaxe Static task.** The timeline shows part of a trajectory where HiSD correctly clusters semantically similar interactions despite varying agent positions, while maintaining clear boundaries between distinct activities.

## D.3. Graphical Results and Discussion of Experiments

We compare the performance of three different skill segmentation and hierarchy induction frameworks: HiSD, OMPN (Lu et al., 2021), and CompILE (Kipf et al., 2019), across a variety of tasks in the Craftax and Minecraft environments. These tasks vary in complexity, planning horizon, and stochasticity. Refer to the main paper for full metrics achieved on each task.

Below we provide some visual examples from a selection of the tasks employed in order to graphically compare the approaches. The combination of quantitative performance and qualitative interpretability across a range of task types illustrates that HiSD is a highly effective approach for skill segmentation and hierarchy discovery. While OMPN and CompILE occasionally perform well in deterministic scenarios, they rely heavily on manual annotation and lack the capacity for semantic skill alignment. CompILE performs reasonably in structured tasks but falls short under noise and longer temporal dependencies. HiSD's flexible, automatic decomposition of skill hierarchies proves particularly valuable in real-world, partially observable domains, such as Minecraft. We note that OMPN's trees must be manually annotated and do not perform any skill matching like HiSD.

- Figure 11 shows results for the *Mixed Static* task. HiSD identifies an extra, erroneous skill segment, seen in its discovered hierarchy (Figure 11C), which does not appear in the ground truth (Figure 11B). However, even though there is an additional incorrect node, we note that this hierarchy is still more informative and interpretable than the one discovered by OMPN, pictured in Figure 11D.

- Figure 12 illustrates the *Stone Pickaxe Random* task. Here, HiSD perfectly recovers the ground truth hierarchy (Figure 12C vs. Figure 12B). In contrast, OMPN's hierarchy (Figure 12D) flattens all actions into a single subtree, lacking any semantic mapping to skill boundaries. These results underscore HiSD's strength in handling noisy, variable-length trajectories.

- Figure 13 presents the *Wood-Stone Collection (Random)* task. All models, HiSD, OMPN, and the ground truth agree on the same decomposition (see Figure 13B–D), suggesting that this task's structure is simple and deterministic enough to allow for consistent interpretation by different methods.

- Figure 14 examines the *Stone Pickaxe Static* task. Although the ground truth (Figure 14B) defines a flat hierarchy, HiSD extracts a more informative tree structure (Figure 14C), revealing alternate sequencing between sub-tasks. Meanwhile, OMPN again falls into the same pattern observed in *Stone Pickaxe Random*, collapsing the hierarchy and introducing unnecessary low-level actions such as redundant walking nodes (Figure 14D).

- Figure 15 visualises the *Minecraft Mapped* task. HiSD produces a concise and generally informative hierarchy (Figure 15A) that, while not matching the ground truth exactly (Figure 15B), captures the overall skill flow and structure effectively. Some incorrect nodes and hierarchy placements are present, but the segmentation is coherent and useful. OMPN's hierarchy is omitted due to its excessive complexity and size, again revealing its limitations in scalability. The segmentation comparison (Figure 15C) further supports HiSD's advantage in structural reasoning.

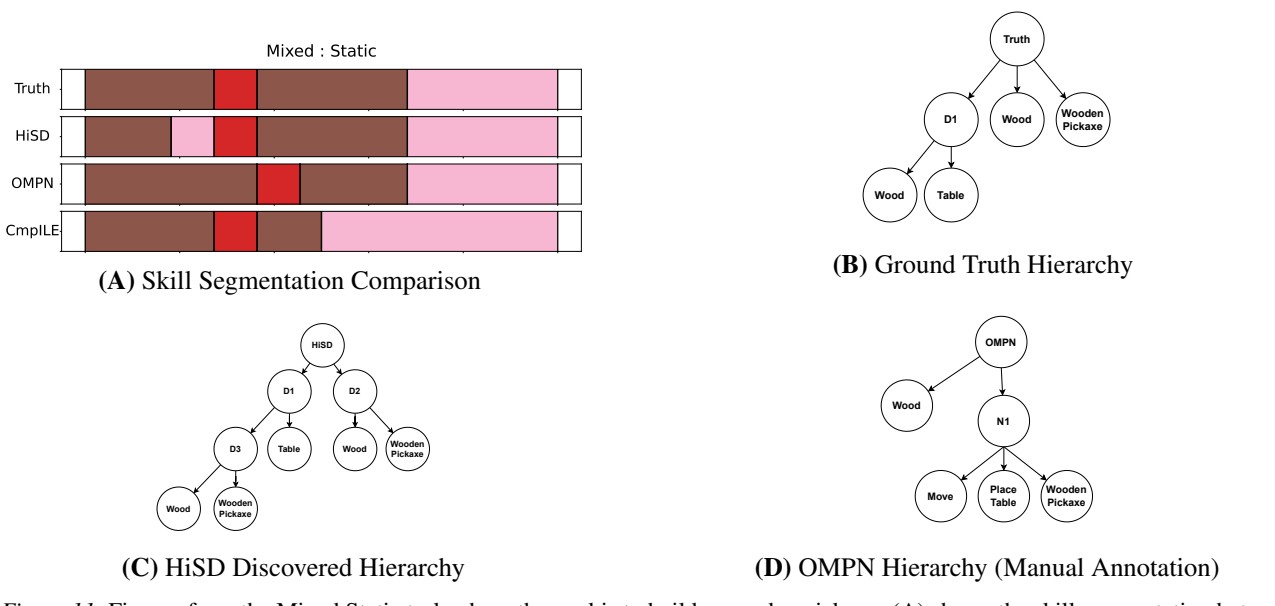

**(A)** Skill Segmentation Comparison

**(B)** Ground Truth Hierarchy

**(C)** HiSD Discovered Hierarchy

**(D)** OMPN Hierarchy (Manual Annotation)

*Figure 11.* Figures from the Mixed Static task where the goal is to build a wooden pickaxe. (A) shows the skill segmentation between baselines. (B) is the ground truth tree, (C) is the tree discovered by HiSD, and (D) is the tree discovered by OMPN.

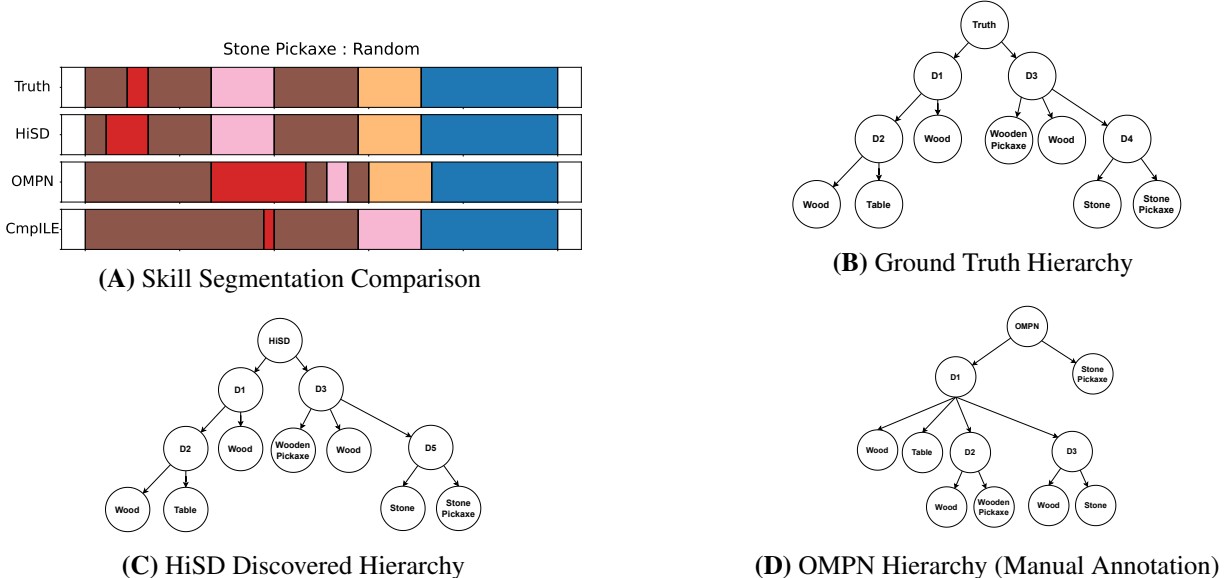

**(A)** Skill Segmentation Comparison

**(B)** Ground Truth Hierarchy

**(C)** HiSD Discovered Hierarchy

**(D)** OMPN Hierarchy (Manual Annotation)

*Figure 12.* Figures from the Stone Pickaxe Random task where the goal is to build a stone pickaxe. (A) shows the skill segmentation between baselines. (B) is the ground truth tree, (C) is the tree discovered by HiSD, and (D) is the tree discovered by OMPN. We see HiSD matches the ground truth exactly, however, OMPN decomposes everything into one subtree (left), with no meaningful connection between subtrees and skills.

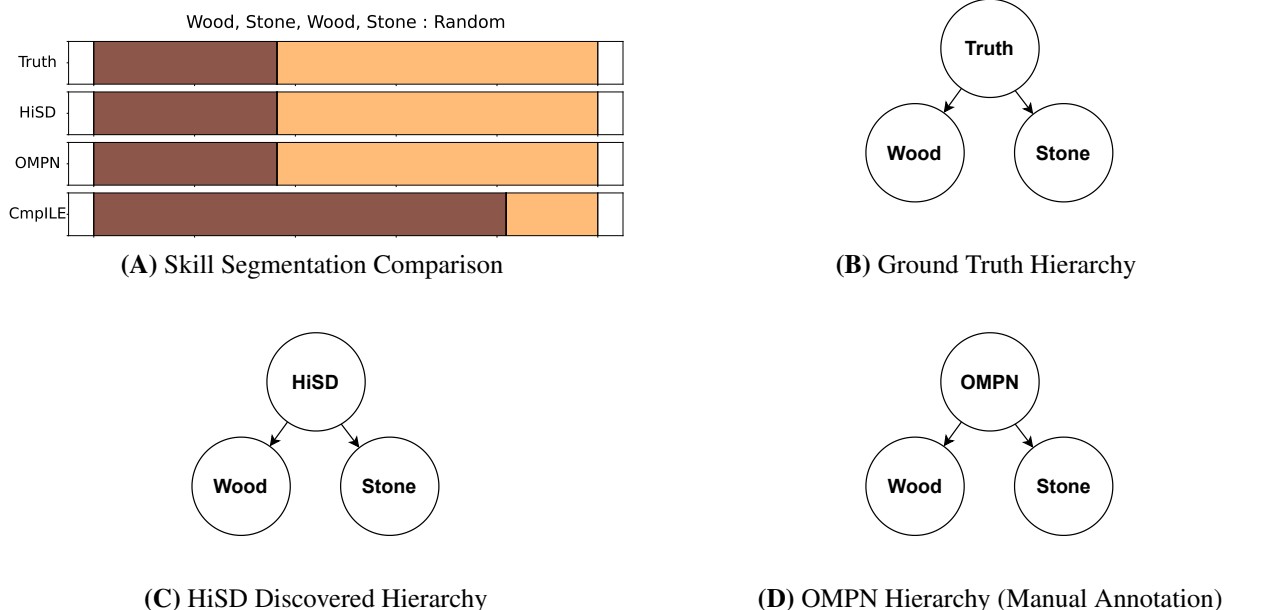

**(A)** Skill Segmentation Comparison

**(B)** Ground Truth Hierarchy

**(C)** HiSD Discovered Hierarchy

**(D)** OMPN Hierarchy (Manual Annotation)

*Figure 13.* Figures from the Wood-Stone Collection (Random) task where the goal in this case is to collect wood and stone, twice each, in any order. (A) shows the skill segmentation between baselines. (B) is the ground truth tree, (C) is the tree discovered by HiSD, and (D) is the tree discovered by OMPN. In this case we see all implementations find the same tree decomposition.

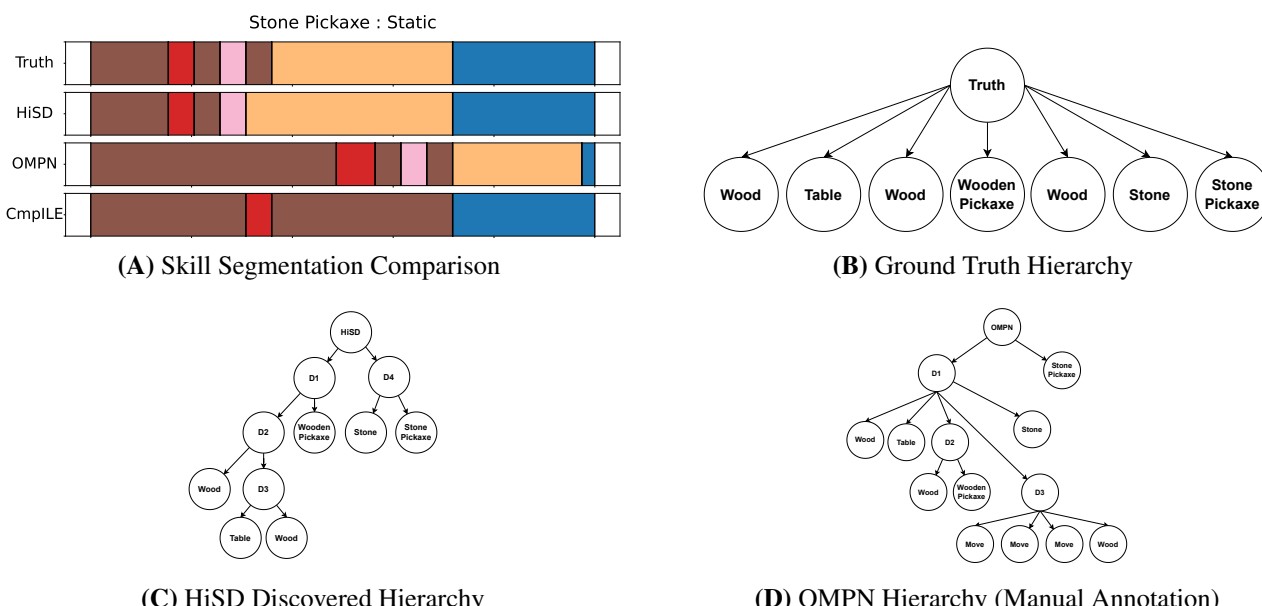

**(A)** Skill Segmentation Comparison

**(B)** Ground Truth Hierarchy

**(C)** HiSD Discovered Hierarchy

**(D)** OMPN Hierarchy (Manual Annotation)

*Figure 14.* Figures from the Stone Pickaxe Static task where the goal in this case is to collect a stone pickaxe. (A) shows the skill segmentation between baselines. (B) is the ground truth tree, (C) is the tree discovered by HiSD, and (D) is the tree discovered by OMPN. In this case, we see that even though the ground truth is a flat hierarchy, due to HiSD finding alternate skill sequencing, it leads to an informative hierarchy. OMPN exhibits the same pattern as in the Stone Pickaxe Random task (Figure 12), where almost the entire decomposition happens under one sub-task, as well as unnecessary nodes being discovered (such as the many walking nodes).

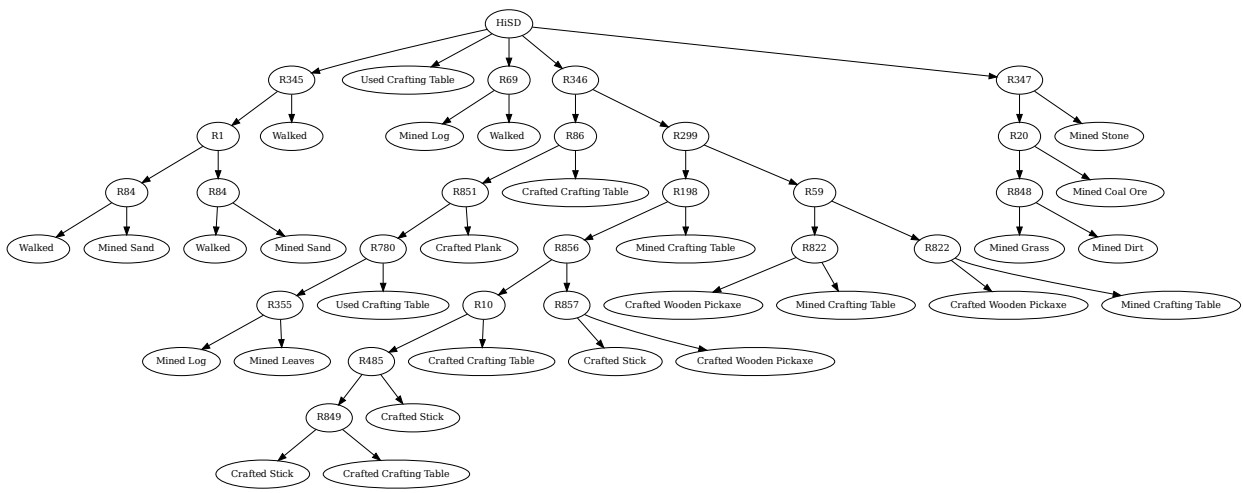

**(A)** Hierarchy Discovered by HiSD for the Minecraft Mapped task.

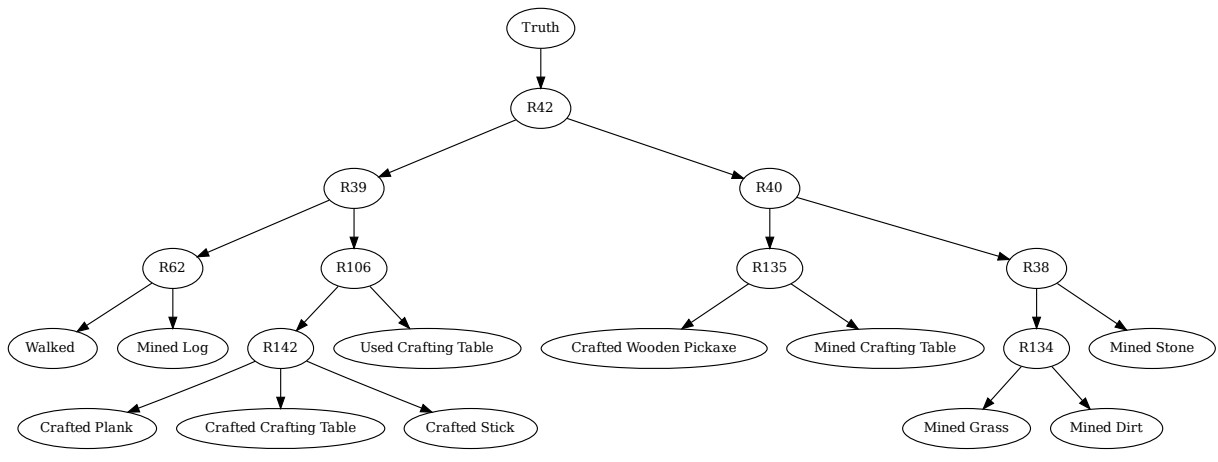

**(B)** Ground Truth Hierarchy for the Minecraft mapped task.

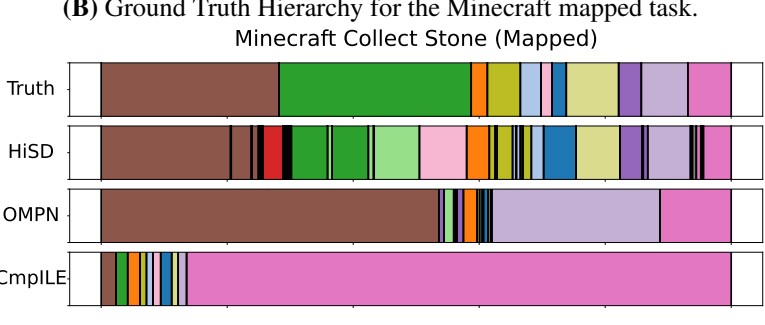

**(C)** Skill Segmentation Comparison

*Figure 15.* Figures from the Minecraft Mapped task where the goal is to collect 2 stone blocks. (A) is the tree discovered by HiSD, (B) is the ground truth tree, and (C) shows the segmentation info. We note that the tree discovered by OMPN is omitted due to its size.

# E. Hyperparameter Selection

We conduct an extensive hyperparameter optimisation for HiSD using the Optuna framework (Akiba et al., 2019). The search space, detailed in Table 18, encompasses both continuous and categorical parameters critical for training stability and evaluation performance. Notably, parameters such as `n-frames` were tuned over broad ranges to accommodate the varying trajectory lengths inherent to Craftax and Minecraft tasks. Furthermore, toggles for upper-bound constraints and feature standardisation were included to ensure HiSD could adapt robustly to different levels of data variability. This comprehensive sweep facilitated strong generalisation across diverse domains.

In contrast, the baselines (OMPN and CompILE) required narrower, predominantly categorical search spaces due to their substantial computational overhead (see Table 16 and Table 17). For OMPN, tuning focused on training dynamics (`il_train_steps`, `il_lr`) and memory slot allocation (`nb_slots`). Similarly, CompILE required careful adjustment of KL regularisation coefficients ($\beta_z, \beta_b$) and prior update rates. Due to extreme resource demands in the Minecraft environment, baseline hyperparameters for those tasks were manually configured based on domain expertise rather than automated sweeping. This disparity in tuning strategies further underscores the computational efficiency of HiSD.

The final selected hyperparameters for HiSD, CompILE, and OMPN across all tasks are reported in Tables 21, 20, and 19, respectively.

| Parameter | Choices |
|---|---|
| `il_train_steps` | {500, 1000, 3000, 5000} |
| `il_lr` | {1e-5, 1e-4, 1e-3, 1e-1} |
| `il_clip` | {0.1, 0.2, 0.3, 0.8} |
| `il_recurrence` | {10, 20, 30, 40} |
| `hidden_size` | {64, 128, 256} |
| `nb_slots` | {2, 3, 4, 5, 6} |

*Table 16.* OMPN hyperparameter search space (Craftax). All parameters are categorical.

| Parameter | Choices |
|---|---|
| `train_steps` | {500, 3000, 5000} |
| `learning_rate` | {1e-4, 1e-3, 1e-1} |
| $\beta_z$ | {0.01, 0.1, 0.5, 1.0} |
| $\beta_b$ | {0.01, 0.1, 0.5, 1.0} |
| `prior_rate` | {3, 5, 10} |
| `hidden_size` | {64, 128, 256} |

*Table 17.* CompILE hyperparameter search space (Craftax). All parameters are categorical.

| Parameter | Type | Range / Choices |
|---|---|---|
| $\alpha$`-train/eval` | Float | $[0.01, 1.0]$ (step 0.01) |
| $\lambda$`-frames-train/eval` | Float | $[0.01, 0.1]$ (step 0.01) |
| $\lambda$`-actions-train/eval` | Float | $[0.01, 0.1]$ (step 0.01) |
| $\epsilon$`-train/eval` | Float | $[0.001, 0.5]$ (step 0.001) |
| `radius-gw` | Float | $[0.001, 0.1]$ (step 0.001) |
| $\rho$ | Float | $[0.001, 0.3]$ (step 0.001) |
| `learning-rate` | Categorical | {1e-5, 1e-4, 1e-3, 1e-2, 1e-1} |
| `weight-decay` | Categorical | {1e-8, …, 1e-1} (log scale) |
| `n-epochs` | Int | $[5, 50]$ (step 5) |
| `ub-frames/actions` | Bool | {True, False} |
| `std-feats` | Bool | {True, False} |
| `n-frames (Craftax)` | Int | $[5, 500]$ (step 5) |
| `n-frames (Minecraft)` | Int | $[100, 30000]$ (step 100) |

*Table 18.* HiSD hyperparameter search space used in the Optuna study. Ranges are continuous unless specified as categorical sets. Task-specific ranges for `n-frames` are noted.

| Task | H-Dim | Clip | LR | Recur. | Slots |
|---|---|---|---|---|---|
| *Craftax* | | | | | |
| WSWS Rand | 64 | 0.8 | 1e-4 | 10 | 2 |
| Stone Stat | 128 | 0.8 | 1e-3 | 20 | 6 |
| Stone Rand | 128 | 0.2 | 1e-3 | 10 | 6 |
| Mixed Stat | 128 | 0.2 | 1e-3 | 40 | 4 |
| *Minecraft* | | | | | |
| All | 128 | 0.8 | 1e-4 | 20 | 8 |
| Mapped | 128 | 0.8 | 1e-4 | 20 | 8 |

*Table 19.* **OMPN** hyperparameters used. Columns: Hidden Size, Clip, LR, Recurrence, Slots.

| Task | $\beta_b$ | $\beta_z$ | LR | Prior | H-Dim |
|---|---|---|---|---|---|
| *Craftax* | | | | | |
| WSWS Rand | 0.01 | 0.1 | 1e-4 | 3 | 128 |
| Stone Stat | 0.1 | 0.01 | 1e-4 | 10 | 256 |
| Stone Rand | 1.0 | 0.1 | 1e-4 | 10 | 256 |
| Mixed Stat | 0.1 | 0.01 | 1e-3 | 3 | 256 |
| *Minecraft* | | | | | |
| All | 0.1 | 0.1 | 1e-4 | 30 | 128 |
| Mapped | 0.1 | 0.1 | 1e-4 | 30 | 128 |

*Table 20.* **CompILE** hyperparameters used. Columns: $\beta_b$, $\beta_z$, LR, Prior Rate, Hidden Size.

| Task | $\alpha$-eval | $\alpha$-train | $\epsilon$-eval | $\epsilon$-train | $\lambda$-action-eval | $\lambda$-action-train | $\lambda$-frame-eval | $\lambda$-frame-train | LR | Epochs | n-frames | GW-Radius | $\rho$ | Std. Feats | UB-Actions | UB-Frames | Weight Decay |
|---|---|---|---|---|---|---|---|---|---|---|---|---|---|---|---|---|---|
| **Craftax** WSWS (Rand) | 0.08 | 0.14 | 0.17 | 0.31 | 0.07 | 0.03 | 0.01 | 0.08 | 1e-4 | 5 | 60 | 0.04 | 0.04 | T | F | F | 0.001 |
| **Craftax** Stone (Static) | 0.14 | 0.11 | 0.38 | 0.02 | 0.10 | 0.10 | 0.03 | 0.10 | 1e-4 | 30 | 135 | 0.10 | 0.18 | T | F | F | 0.001 |
| **Craftax** Stone (Rand) | 0.22 | 0.59 | 0.02 | 0.01 | 0.09 | 0.06 | 0.10 | 0.10 | 0.01 | 5 | 110 | 0.01 | 0.12 | F | F | T | 0.01 |
| **Craftax** Mixed (Static) | 0.21 | 0.03 | 0.05 | 0.001 | 0.10 | 0.09 | 0.01 | 0.01 | 1e-4 | 50 | 205 | 0.09 | 0.01 | F | T | T | 0.001 |
| **Minecraft** All | 0.60 | 0.66 | 0.44 | 0.08 | 0.06 | 0.10 | 0.08 | 0.06 | 1e-4 | 45 | 1800 | 0.01 | 0.12 | T | F | F | 0.1 |
| **Minecraft** Mapped | 0.95 | 0.23 | 0.34 | 0.07 | 0.01 | 0.03 | 0.02 | 0.07 | 1e-4 | 30 | 9100 | 0.01 | 0.16 | F | F | F | 0 |

*Table 21.* Final hyperparameters for HiSD Skill Segmentation.

