# OpenReview forum: "Unsupervised Hierarchical Skill Discovery"
_ICML.cc/2026/Conference — ICML 2026 regular_

### Official Review · Reviewer_v1FB · 2026-03-07

**Soundness:** 2
**Presentation:** 2
**Significance:** 3
**Originality:** 3
**Overall Recommendation:** 3
**Confidence:** 4

**Summary:**

- This paper focuses on learning a hierarchical set of skills from an offline dataset without any supervision.
- The proposed method, HiSD, first splits given demonstrations into skill-level segments, and then compresses the resulting skill sequence into a grammar-based sequence.
- In the Craftax and Minecraft environments, the method shows strong performance on both skill segmentation and agent control.

**Compliance With Llm Reviewing Policy:**

Affirmed.

**Final Justification:**

Thank you for the authors’ detailed responses. I understand that the natural-language skill matching is a secondary aspect of the work. However, considering the paper as a whole and the authors’ contributions, I will maintain my original score.

**Key Questions For Authors:**

- Could the authors explain the ASOT process in more detail, since HiSD relies on ASOT during training?
- How are ASOT’s discrete skills converted into natural-language skill descriptions (as shown in Figure 3)?

**Limitations:**

Please refer to the “Weaknesses” part.

**Strengths And Weaknesses:**

### Strengths
- HiSD presents an interesting idea. It simplifies demonstrations into sequences of skills by combining ASOT with grammar-based sequence compression.
- The paper uses a wide range of metrics to compare and analyze the quality of the learned skill segmentations.
- Through reinforcement learning experiments, the paper shows that it is possible to connect hierarchical tree construction with low-level action prediction.

### Weaknesses
- The paper title and the method name (HiSD) feel too broad and do not reflect the author’s main idea well. Also, the overall description of the proposed method is not very clear, which makes it hard to fully understand how the method works.
- The paper does not explain ASOT in enough detail. HiSD uses ASOT as a key component (to split a demonstration into multiple segments), but it is unclear what signals or features ASOT uses to produce each segmentation. In addition, Eq. (1) includes some functions and arguments that are not defined or explained in the text.
- Each skill $z$ seems to be defined as a discrete skill label, but Figure 3 describes the segmentation result as a natural-language skill sequence. The paper does not clearly explain how the index of $z$ is mapped to natural-language descriptions.
- Since there are many prior works that learn hierarchical skill from offline datasets, more experiments with relevant baselines seem necessary.

---

> ### Author Rebuttal · Authors · 2026-03-29
>
> We thank the reviewer for acknowledging the breadth of our evaluation metrics and the connection to downstream RL. We address each concern below:
>
> **W1: The title and method name feel too broad:**
> We appreciate this feedback and are open to a more specific title. That said, each word maps to a demonstrated property:
> - Unsupervised: No action labels, reward signals, annotations, or online interaction during discovery (Sections 1, 4).
> - Hierarchical: Multi-level compositional structure via grammar-based compression, yielding tree-structured decompositions with internal nodes as discovered subroutines (Sections 2.2, 4.2; Figures 11-15).
> - Skill Discovery: Discrete, reusable behavioural units identified from continuous observation streams via temporal action segmentation (Sections 2.1, 4.1; Table 1). Our use of "skill" is consistent with established terminology where a skill denotes a temporally extended, reusable behavioural primitive (e.g., NPBRS, Ranchod et al., 2015; CompILE, Kipf et al., 2019; OMPN, Lu et al., 2021).
>
> **W2: ASOT is not explained in enough detail; Eq. 1 contains undefined elements:**
> Thanks for flagging this. While $\mathbf{C} \in \mathbb{R}^{n \times K}$ (cost matrix between observations and skill prototypes) and $\boldsymbol{\Gamma} \in \mathbb{R}^{n \times K}_+$ (soft assignment plan) are introduced in the surrounding text, the following were insufficiently defined and will be clarified:
> - $\eta > 0$: weight on the temporal regularity term $R_{\mathrm{temp}}$.
> - $\lambda > 0$: weight on the KL-divergence penalty, softly constraining $\boldsymbol{\Gamma}$'s column-sum marginal towards a target $\mathbf{q}$.
> - $\mathbf{q} \in \mathbb{R}^K$: the target marginal over skills (set to uniform, $\mathbf{q} = \frac{1}{K}\mathbf{1}_K$).
> - $\mathbf{1}_n \in \mathbb{R}^n$: the $n$-dimensional ones vector.
>
> $R_{\mathrm{temp}}$ is a Gromov-Wasserstein term encoding temporal consistency:
> $$R_{\mathrm{temp}}(\boldsymbol{\Gamma}) = \langle \mathbf{C}^v \boldsymbol{\Gamma} \mathbf{C}^a, \boldsymbol{\Gamma} \rangle$$ where $\mathbf{C}^v \in \mathbb{R}^{n \times n}$ encodes temporal proximity between frames, with $C^v_{ik} = 1/r$ if $1 \leq |i - k| \leq nr$ (for radius parameter $r \in [0,1]$) and $0$ otherwise; and $\mathbf{C}^a \in \mathbb{R}^{K \times K}$ encodes skill identity, with $C^a_{jl} = 0$ if $j = l$ and $1$ otherwise. This penalises plans mapping temporally adjacent frames to different prototypes, encouraging smooth, contiguous segments.
> HiSD is agnostic to the specific segmentation algorithm; ASOT is one instantiation (Section 4.1, line 155). We will expand the ASOT description and add a self-contained mathematical appendix covering the full formulation, parameter–hyperparameter mapping (Table 16), and solver details.
>
> **W3: Figure 3 shows natural-language skill names, but the model outputs discrete labels:**
> HiSD does not produce natural-language descriptions at any stage. As stated in Section 2.1 (line 67) and Section 4.1 (line 160), the model outputs discrete integer labels $z_t \in \{1, \ldots, K\}$. The human-readable names shown in Figure 3 ("wood," "table," etc.) are post-hoc annotations applied for visualisation: we use Hungarian matching to align predicted clusters with ground-truth identities, then replace integers with matched names. This is standard practice in unsupervised segmentation evaluation; Xu & Gould (2024) use the same protocol, as do Ding et al. (2023), Li & Todorovic (2021), CompILE, and OMPN. We will clarify this.
>
> **W4: More baselines from prior work on hierarchical skill discovery from offline datasets:**
> We would be grateful if the reviewer could identify specific works. To our knowledge, no existing method simultaneously: (1) learns hierarchical (not flat) structure, (2) from offline data, and (3) without requiring action labels or reward signals. Specifically:
> - DUSDi, DIAYN, and SkiLD learn skills through online environment interaction, not from offline data.
> - SPiRL and Skill-Critic discover skills offline but require action labels and produce flat decompositions.
> - LOKI performs offline skill discovery but requires action labels and weak task supervision.
> - SBD (Zhu et al., 2022) requires action labels and does not induce hierarchy.
> - HTN-based methods (HTN-Maker, Circuit-HTN, CurricuLAMA) produce hierarchies but require heavily annotated symbolic input.
>
> OMPN and CompILE are the closest comparisons, and we further advantaged both with ground-truth skill orderings at inference that HiSD does not require. We welcome specific suggestions of methods meeting these criteria and would readily include additional comparisons in revision.
>
> **Minor Points:**
> **Could the authors explain ASOT in more detail?**: Yes, as detailed in W2, we will add a self-contained ASOT appendix with full definitions.
> **How are discrete skills converted to natural-language descriptions?**: As noted in W3, the names are post-hoc visualisation labels, not model outputs.

---

> > ### Author Rebuttal · Reviewer_v1FB · 2026-04-01
> >
> > I have carefully reviewed the authors’ response, and many of my concerns have been addressed. However, I still have some remaining questions.
> >
> > Regarding W3, I already understood that the model outputs discrete integers, but I was unsure how these outputs are converted into natural language.
> > The authors explain that they align integers and language using Hungarian matching, but the proposed approach does not appear to include any auxiliary term that encourages a one-to-one mapping between each integer and a single language label.
> > In other words, one integer could potentially map to multiple skills, and multiple skills could also map to a single integer.
> > As such, it is not fully convincing to claim that Hungarian matching alone can reliably translate discrete actions into language.
> >
> > Finally, I will revise my score to 3.

---

> > > ### Author Response · Authors · 2026-04-02
> > >
> > > We thank the reviewer for their continued engagement and for raising this point. We believe there may be a misunderstanding that we are happy to clarify.
> > >
> > > Hungarian matching is not part of the HiSD pipeline. It plays no role during training, inference, or skill discovery. HiSD outputs discrete integer labels and requires no mapping to natural language at any stage. The Hungarian matching is applied _solely for evaluation purposes_, specifically to compute standard metrics (mIoU, F1, MoF) that require a correspondence between predicted cluster indices and ground-truth label indices. This is identical to the evaluation protocol used across the unsupervised segmentation literature (e.g., Xu & Gould, 2024; Ding et al., 2023; Li & Todorovic, 2021), where predicted and ground-truth label sets are aligned post hoc to enable quantitative scoring.
> > >
> > > To clarify the reviewer's specific concern: the reviewer notes that "one integer could potentially map to multiple skills, and multiple skills could also map to a single integer." This is precisely what the metrics are designed to detect and penalise. When the predicted clustering does not achieve a clean one-to-one correspondence with ground-truth skills, metrics such as mIoU and F1 will reflect this through lower scores. The Hungarian matching finds the best possible alignment to give the model the benefit of the doubt during scoring. It does not assert or enforce that the mapping is perfect. Again, we note that this is the standard evaluation approach in unsupervised segmentation.
> > >
> > > The natural-language labels in Figure 3 (e.g., "wood," "table") are simply the ground-truth names substituted in place of the matched integers for the reader's convenience. Removing these labels entirely and displaying raw integers (e.g., Skill 0, Skill 3, Skill 1, ...) would not change any result, only readability. We will revise Figure 3 and the surrounding text to make this evaluation-only role explicit and prevent further confusion.

---

### Official Review · Reviewer_DCST · 2026-03-10

**Soundness:** 3
**Presentation:** 3
**Significance:** 3
**Originality:** 2
**Overall Recommendation:** 4
**Confidence:** 5

**Summary:**

This work addresses the problem of discovering modular and reusable skills and inferring the hierarchical structure composed of these atomic skills. The method proceeds in a two stage manner, first it segments trajectories of observation-only data into segments using the ASOT algorithm for clustering sequence features into K discrete segment labels. Then it compresses the sequence of segmented labels (atoms) into a hierarchical tree using Grammar induction, specifically the Sequitur algorithm. A main difference in their formulation compared to prior works is they only rely on observations to generate the skill segments unlike prior works which use both observations, actions (and rewards). Results show superior grouping performance on and compared to baselines (OMPN and CompILE).

**Compliance With Llm Reviewing Policy:**

Affirmed.

**Final Justification:**

The authors responses to my questions, their responses have helped me better understand the motivations behind various aspects of their work.

Regarding, the end-to-end vs modular design in HiSD, I recognise the authors argument about the fact that prior work made the hierarchical inference problem easier by using additional information about the problem structure available to the learner. And it is commendable that the HiSD shows strong performance without making many of these assumptions about the problem structure compared to prior work.

I lean towards accepting the paper with my original score of 4.

**Key Questions For Authors:**

From my perspective, it is reasonable to define a skill as a compressed (latent) representation z of a sequence of actions taken to achieve a sub-goal from a given initial state, z = f(o_<t, a_<t). So this paper’s formulation of choosing to simply group a sequence of observations feels counterintuitive. I see the authors argument that the this choice allows their algorithm to be used on unlabelled videos, but isn’t the value of learning the compressed latents (observations, actions), that it allows us to train a decoder \pi(a | o, z) that decompresses these high-level skill latents into primitive actions. High-level planning/exploration algorithms can simply operate at the level of these latent skills z. It seems plausible that there is high visual similarity between o_t and o_t+1 (most pixels/features remain invariant to different actions) even though we have taken two different actions a_t or a’_t at time t that achieve different sub-goals. So the fact that skill latents only encode purely chunks of observations and doesn’t contain any trace of actions seems very counterintuitive, given that later for downstream tasks the actions are reintroduced to learn policies by behavioral cloning.

In Table 2, the number of unique trees discovered by both HiSD and OMPN are consistently much larger than the number of ground-truth ones except on the simplest WSWS environment. While I appreciate the redundancies are fewer for HiSD compared to OMPN, given the HiSD uses a targeted method for specifically inferring hierarchies, I would have expected fewer redundant trees? The authors suggest that this is due to compounding of errors in the segmentations from stage 1, doesn’t this hint at a broader point that the algorithm used for learning hierarchies also be robust to noise and not entirely deterministic. While the end-to-end learned baseline OMPN doesn’t fare well either, at least in-principle it has the possibility to error correct from noisy prefixes observed so far. The authors could be more explicit about the failure mode of their fully-deterministic grammar induction algorithm on noisy sequences, possibly by using some probabilistic grammars instead?

In Minecraft, how sensitive are the baselines (OMPN, CompILE) to the action discretization preprocessing that results in 2385 actions? How do they perform if this is more coarsely discretized, i.e. fewer actions?





Comments:

I didn’t quite get the motivation to distinguish between per-episode vs full-dataset metrics for F1 and mIoU. Shouldn’t the full dataset F1 score as the average F1 score across all episodes suffice? And in that case, why not simply report the full dataset F1 score.

How are the feature embeddings extracted for each trajectory given the frames which are then used by the ASOT algorithm? Authors simply say their HiSD algorithm “operates on pre-extracted feature representations”. Given the earlier comment about the sensitivity of the hierarchy induction on the learned atomic segments, it is plausible that if the initial clustering algorithm used “better” features it could alleviate the brittleness to noise or learning features along with the clustering assignments in an end-to-end way.

Prior work [1] has extended CompILE and OMPN to the setting where it only requires a prior over the number of skills K in a given dataset and no need for any other supervision signals. This work relies on an end-to-end learned clustering algorithm that enforces temporal contiguity (smoothness) and modularity of segments modelled by each slot. It also showed the limits and failure modes of these baselines (CompILE and OMPN) when used on sequences with a variable number of sub-routines per sequence in MiniGrid environments.

[1] Unsupervised Learning of Temporal Abstractions With Slot-Based Transformers. Gopalakrishnan et. al, Neural Computation 2023; 35 (4): 593–626.

**Limitations:**

yes.

**Strengths And Weaknesses:**

The paper is very well-written and clearly outlines the research question, method and experimental setup (baselines, environments) is in-line with prior work in the area.

The performance gains on episode-level and dataset-level grouping of skill segments for HiSD are pretty consistent over the baselines CompILE and OMPN on several environments/tasks in Craftax and Minecraft.

The RL deployment experiment also shows that the quality of the learned skill segments speeds up learning on downstream tasks compared to CompILE and OMPN and helps assess the utility of these skill segments beyond just their segmentation quality.

The proposed method HiSD is rather heavily engineered with specialized algorithms for each component of the whole hierarchical structure inference problem. Compared to baselines like OMPN which is simply a hierarchical RNN, that uses unsupervised trajectory level reconstruction to infer all these hierarchical latents in an end-to-end manner with gradient descent.

---

> ### Author Rebuttal · Authors · 2026-03-29
>
> We thank the reviewer for their thorough review and recognition of the paper's clarity, consistent performance gains, and the value of the RL deployment experiments.
>
> **HiSD is heavily engineered with specialised algorithms versus end-to-end baselines:**
> We see HiSD's modularity as a natural consequence of weaker supervision, not an arbitrary design choice. End-to-end baselines appear simpler because the problem they solve is already heavily constrained: OMPN needs action labels, ground-truth skill orderings, and a predefined hierarchy depth; CompILE needs action labels, known segment counts, and a prior over segment length. These signals do the heavy lifting that lets a single network succeed. HiSD requires only a loose upper bound K and observations alone. Modularity also confers practical advantages: components can be independently improved (section 4.1), the framework runs ∼53× faster than CompILE (37s vs. 1,952s per run on Craftax), and operates on consumer GPUs (6 GB VRAM) vs. 24 GB GPUs with 128 GB RAM for baselines (Table 10).
>
> **Key Questions**
> * **Q1 (Observation-only skills and visual similarity between consecutive frames):** Our segmentation does not rely on frame-by-frame visual clustering alone. ASOT encodes a temporal consistency prior via its GW component (R_temp in Eq. 1), penalising label changes between adjacent frames unless the cost structure supports a transition, enforcing stable labels over coherent behavioural segments. Figure 10 shows HiSD learns object-centric skills robust to positional variance. At deployment, segmentation groups frames into semantically meaningful chunks (e.g., "mining wood"). The BC policy $\pi_i(a|s)$ within each segment learns the action distribution conditioned on observation and skill context: the skill label specifies what to do; BC learns how. Two visually similar states in different skills (walking-to-wood vs. walking-to-stone) receive different BC policies trained on appropriate actions. RL results (Figures 5 and 6) confirm this works in practice.
>
> * **Q2 (Unique tree counts and deterministic grammar):** In the Minecraft "All" setting (44 skills, noisy VPT demos), inconsistencies produce variable sequences preventing convergence to few trees. When noise is reduced (Mapped: 14 skills, or Craftax), counts approach ground truth closely (HiSD: 9 vs. OMPN: 499 for WSWS Random). Deterministic grammars offer interpretability (hierarchies can be directly inspected), but we will add probabilistic grammar extensions as explicit future work.
>
> * **Q3 (Baseline sensitivity to action discretisation):** We did not ablate coarser discretisations. Collecting all unique observed actions (2,385 integers) is the most assumption-free mapping; any coarser scheme would require domain-specific merging that could inadvertently advantage or disadvantage particular methods.
>
> * **Q4 (Per-episode vs. full-dataset metrics):** Per-episode (Per) matching lets each episode independently align clusters with ground-truth skills, assessing local quality. Full-dataset (Full) enforces a single global alignment across episodes. Example: if cluster 3 maps to "wood" in episode A but "stone" in episode B, Per scores both correctly while Full penalises the inconsistency. We prioritise Full mIoU because cross-episode consistency is essential for downstream hierarchical RL, where skills must transfer across contexts. We will clarify this in revision.
>
> * **Q5 (Feature embeddings and end-to-end learning):** In Craftax, we apply PCA (650 components, ∼99% variance) to raw 274×274×3 RGB observations. In Minecraft, we use frozen MineCLIP embeddings (512-d per frame). Both are fixed preprocessing; ASOT and grammar induction operate downstream. Jointly learning features with segmentation could improve performance (noted in section 8); ASOT achieves SoTA with end-to-end features (Xu & Gould, 2024), though this would add complexity that may offset modularity advantages.
>
> * **Q6 (SloTTAr reference):** We thank the reviewer for highlighting this work. SloTTAr (Gopalakrishnan et al., 2023) extends CompILE/OMPN with slot-based Transformers that handle variable subroutine counts. While the reviewer notes it removes the need for other supervision signals (e.g., ground-truth segment counts per episode), SloTTAr still operates on state-action trajectories: its architecture attends to both observation and action features and reconstructs action sequences via its decoder. It was evaluated on simple grid-worlds (Craft, MiniGrid), and produces flat segmentations without hierarchical structure. In contrast, HiSD operates on observations alone (no actions at any stage of discovery) and induces multi-level hierarchies. We will cite this work in the revised manuscript and discuss its relationship to our approach in the skill learning phase.

---

> > ### Author Rebuttal · Reviewer_DCST · 2026-04-01
> >
> > I appreciate the authors responses to my questions, their responses have helped me better understand the motivations behind their model design (modular approach with lower compute/memory requirements, deterministic grammars, frozen features), certain results (unique tree counts), evaluation criteria (per-episode vs full).
> >
> > The broader point I was trying to get at with the hand-crafted design choices in each of the modules can also lead to increased complexity via extra hyperparameters or modelling assumptions, interfaces between these modules etc. Allowing learning to maximally influence all modules of the hierarchical skill discovery system to learn suitable low-level features, groupings, tree inference would make the system-level recipe more portable to a new environment and learn hierarchies without needing many interventions (design choices) from the human designer.
> >
> > Regarding, the end-to-end vs modular design in HiSD, I recognize the authors argument about the fact that prior work made the hierarchical inference problem easier by using additional information about the problem structure available to the learner. And it is commendable that the HiSD shows strong performance without making many of these assumptions about the problem structure compared to prior work.
> >
> > I'm happy to maintain my score at 4.

---

> > > ### Author Response · Authors · 2026-04-02
> > >
> > > Thank you for your thorough and insightful review. Your questions around modularity, end-to-end learning, and evaluation design pushed us to articulate our design rationale more clearly, and we are grateful for the discussion.

---

### Official Review · Reviewer_cSgz · 2026-03-11

**Soundness:** 2
**Presentation:** 2
**Significance:** 2
**Originality:** 3
**Overall Recommendation:** 4
**Confidence:** 3

**Summary:**

This paper proposes HiSD, a fully unsupervised framework that extracts reusable, multi-level skill hierarchies from observational data. The authors claim that the method can learn both skills and their hierarchical structure without requiring reward signals or action supervision, which are typically needed in prior approaches. Experiments conducted in the Craftax and Minecraft environments demonstrate that HiSD produces more structured and semantically meaningful hierarchies compared to existing methods.

**Compliance With Llm Reviewing Policy:**

Affirmed.

**Final Justification:**

The rebuttal sufficiently addressed my main concerns, particularly regarding scalability to more complex environments and the supervision available to the baselines, as well as my other comments. Accordingly, I have revised my score to 4.

**Key Questions For Authors:**

1. In the Craftax environment, feature representations are obtained using PCA, whereas MineCLIP is used in the Minecraft environment. What is the reason for using different feature extraction methods across these environments?
2. In Section 6.1.1, the paper states: “in Craftax and Minecraft, visually distinct approaches to the same object (such as collecting wood from different directions or angles) are consistently assigned to the same skill cluster.” Given that HiSD uses only observations without action information, what factors enable the model to achieve this behavior?
3. In the experiments, are reward signals and action labels provided to CompILE and OMPN?
4. If reward and action data were also provided to HiSD, would its performance improve compared to the observation-only setting?

**Limitations:**

Yes.

**Strengths And Weaknesses:**

**Strengths**

- Applying the ASOT framework and the Sequitur method, which have previously been used in other domains, to the problem of skill discovery appears to be a key aspect that distinguishes this work from prior studies.
- The experimental design and evaluation metrics effectively highlight the strengths of HiSD.

**Weaknesses**

- While learning skills without reward signals or action labels is a clear strength, it remains unclear how well this observation-only setting would extend to more complex environments. HiSD relies on ASOT-based skill segmentation, which depends mainly on state similarity and temporal consistency. As a result, segmentation may become challenging when different skills occur in similar states or when the same skill appears under diverse observations. Since the evaluated environments, Craftax and Minecraft, mainly involve mining and crafting tasks where skills and states are relatively well aligned, segmentation based solely on observations may be easier in these settings. Evaluating the method in environments with more varied interactions would provide a clearer picture of the robustness of HiSD.

- The Sequitur-based grammar effectively captures frequently occurring skill combinations and their sequential structure, but it has limitations in terms of causal interpretation. It primarily learns a compressed representation of commonly observed skill sequences rather than identifying whether certain skills are necessary preconditions for others. As a result, even when sequential patterns appear similar, the learned hierarchy may group them into the same subroutine despite differences in their underlying constraints. This limitation could reduce the grammar’s usefulness in environments that require reasoning about preconditions.

---

> ### Author Rebuttal · Authors · 2026-03-29
>
> We thank the reviewer for recognising the originality of our approach and the effectiveness of our experimental design. We address each concern below:
>
> **W1: Observation-only segmentation may not extend to more complex environments:**
> We respectfully disagree with the characterisation that Craftax and Minecraft are environments where skills and states are well-aligned and thus unrepresentative of complex settings. As highlighted by reviewer DCST, our "experimental setup (baselines, environments) is in-line with prior work in the area." Our evaluation domains are substantially more challenging than those of our baselines: CompILE and OMPN were evaluated on toy integer sequences, simple grid-worlds, and low-dimensional continuous control tasks. In contrast, Craftax and Minecraft use high-dimensional pixel observations where different skills can occur in visually similar states.
>
> In Minecraft, the "walked" skill alone accounts for ~6% of the total frames in the Mapped dataset (Figure 9b). Walking-to-wood and walking-to-stone are observationally indistinguishable from first-person pixels, yet correspond to different goals. To illustrate, we provide a visual example of the ambiguity between walking to wood and walking to stone; this observation could correspond to either skill ([image linked here](https://i.imgur.com/5kK5kmn.jpeg)). The crafting GUI presents a similar challenge: crafting planks, sticks, and a pickaxe all within the same visual interface. Unlike simplified Minecraft frameworks where crafting is treated as a discrete action (Kanervisto et al., 2020), we use the full, native GUI, requiring the agent to visually parse an identical-looking interface to distinguish different crafting skills. Overall, we estimate over 25% of Mapped frames involve observationally ambiguous states. Despite this, HiSD achieves 38% Avg. mIoU on Mapped, outperforming OMPN (14%) and CompILE (6%) by large margins, even though both baselines receive action labels and ground-truth skill orderings that HiSD does not. In Craftax, Figure 10 and Appendix D.2 show that HiSD correctly distinguishes semantically distinct skills in visually similar states (e.g., "Table" interaction vs. "Wood" gathering when the workbench is adjacent to trees).
>
> **W2: Sequitur captures sequential co-occurrence rather than causal preconditions:**
> We agree that Sequitur learns compressed representations of co-occurring skill sequences rather than explicit causal preconditions, which we view as an orthogonal problem rather than a limitation of skill discovery per se. However, this limitation is shared by all compared methods: neither CompILE, OMPN, NPBRS, nor grammar-based methods (Lange & Faisal, 2019) model causal structure. All rely on statistical regularities or assumed structure. Importantly, despite this shared limitation, HiSD's hierarchies are empirically the most useful for downstream RL (Figures 5 and 6), suggesting that the statistical regularities captured are practically meaningful. We will add causal grammar induction explicitly as a future work direction in the revised manuscript.
>
> **Key Questions**
> * **Q1 (Different feature extractors):** Craftax is fully observable; PCA retains ~99% variance and is computationally sufficient. Minecraft generates 640×360 first-person frames in a partially observable, stochastic environment, making PCA both computationally prohibitive and semantically inadequate. MineCLIP provides pre-trained 512d embeddings capturing visual semantics. HiSD is agnostic to the choice of feature extractor (Section 4.1); the algorithm itself is unchanged across domains.
> * **Q2 (Grouping visually distinct views):** Two complementary factors enable this: (1) feature representations capture semantic similarity, so different views of the same object map to nearby points in feature space; and (2) ASOT's temporal consistency regularisation ($\mathcal{R}_{\mathrm{temp}}$ in Eq. 1) penalises assigning adjacent frames to different labels unless the cost structure warrants it, enforcing smooth labels over coherent temporal segments.
> * **Q3 (Supervision for baselines):** Yes, both CompILE and OMPN require action labels during training, and they do not function without them. At inference, both also receive ground-truth sub-task orderings. CompILE further requires segment count and average skill length; OMPN requires a predefined hierarchy depth. Despite this strictly greater supervision, both are consistently outperformed by HiSD on more complex tasks (Table 1). None of the baselines used require reward signals.
> * **Q4 (Would incorporating actions improve HiSD?):** Appending discretised actions yielded mixed results: +6.3% Avg. mIoU on Stone Pickaxe Static but −10.2% on Random. Actions can introduce noise that disrupts temporal coherence, particularly where the same skill involves different action sequences. The observation-only setting provides robustness to action variability - a valuable property for imperfect demonstrations.

---

> > ### Author Rebuttal · Reviewer_cSgz · 2026-04-01
> >
> > Thank you for the detailed response. After reviewing your rebuttal, I find that most of my concerns have been sufficiently addressed. In particular, I found your clarification regarding whether observation-only segmentation can scale to more complex environments to be the most compelling. In addition, the fact that the comparison methods require stronger supervision further clarifies the practical strengths of HiSD. Based on these responses, I will revise my score to 4.

---

> > > ### Author Response · Authors · 2026-04-02
> > >
> > > Thank you for your continued engagement and for revisiting your assessment. Your questions helped us better articulate the strengths of the observation-only setting, and we appreciate the constructive dialogue.

---

### Official Review · Reviewer_74Qm · 2026-03-12

**Soundness:** 3
**Presentation:** 2
**Significance:** 2
**Originality:** 3
**Overall Recommendation:** 4
**Confidence:** 3

**Summary:**

This paper proposes an unsupervised method for discovering skills and organizing them into a hierarchical structure from observational trajectories. The approach first segments trajectories into latent skills, and then applies grammar-based sequence compression to induce hierarchical structure over the discovered skills. The authors demonstrate how these skills can be used as options within a hierarchical reinforcement learning framework, where a high-level policy selects among these skills.

The method is evaluated in two environments (Craftax and Minecraft) using metrics for both skill segmentation quality and hierarchy structure. The authors additionally demonstrate that the learned hierarchies can improve the sample efficiency of downstream reinforcement learning compared to non-hierarchical policies.

**Compliance With Llm Reviewing Policy:**

Affirmed.

**Final Justification:**

The rebuttal addressed my main concerns and included an additional experiment on few-label supervision. The overall contribution is solid. I maintain my score of 4.

**Key Questions For Authors:**

1. The paper highlights that the method requires prior knowledge of the maximum number of skills K. Is there a way to infer K automatically from the data?
2. Line 849 mentions training duration as “slow” and “fast”. Could you provide concrete numbers for these training times?
3. Since the proposed method is more computationally efficient than the baselines, could the performance improvements partially stem from allowing more extensive hyperparameter tuning?
4. I did not fully understand the difference between fixed and stochastic task ordering. Could you clarify how these settings differ and how they affect the evaluation?
5. The paper would benefit from a clearer description of the datasets used in the experiments, including the final dataset sizes. In particular, could you clarify the construction and size of the "all-skills" and "mapped" datasets?

**Limitations:**

The paper includes a brief discussion of the limitations of the method. It would be beneficial to expand this discussion further, particularly regarding how the method might generalize to real-world observational datasets and domains with overlapping or concurrent skills.

**Strengths And Weaknesses:**

**Strengths**

1. The paper addresses an important challenge in reinforcement learning: discovering reusable hierarchical structure directly from demonstrations without relying on action labels or reward signals. This setting is particularly relevant for large-scale observational data such as human demonstrations.
2. The proposed approach is conceptually simple: it combines temporal segmentation of trajectories with grammar-based hierarchy induction in a straightforward pipeline. The related work section clearly positions the approach and explains how it differs from existing methods.
3. The approach appears to be technically sound. The authors evaluate their method using two types of metrics (segmentation quality and hierarchical structure quality) and two environments: simplified Craftax and the Minecraft environment with imperfect demonstrations. The method is compared against several baselines. The authors also demonstrate improvements on downstream reinforcement learning tasks ("Craft Wooden Pickaxe" in Craftax and a relatively simple "Collect Log" task in Minecraft).
4. The appendix includes visualizations of the learned hierarchies and examples of failure cases, which helps illustrate how the method behaves.

**Weaknesses**

1. The overall significance seems somewhat limited. While the method is described as unsupervised, the downstream RL experiments demonstrate improvements only for behavior cloning with action labels. It would be particularly interesting to evaluate the method in settings with behavior cloning without action labels or with only a small amount of action labels, as well as in robotic domains, where unsupervised skill discovery could be especially valuable.
2. The proposed pipeline assumes that each timestep is assigned to a single discrete skill. While this assumption is reasonable in domains where behaviours occur sequentially (such as the Craftax tasks used in the experiments), it may be restrictive in real-world robotic settings where multiple behaviours can occur concurrently (for example, reaching while stabilizing an object or coordinating arm and gripper motions). In such cases, forcing trajectories into a single discrete skill sequence may lead to unstable segmentations and reduce the effectiveness of the grammar-based hierarchy induction stage. It would be interesting to understand how the proposed method behaves in domains with overlapping or concurrent skills.

---

> ### Author Rebuttal · Authors · 2026-03-29
>
> We are grateful for the reviewer's recognition of our method's conceptual simplicity, technical soundness, and comprehensive evaluation across both segmentation and hierarchy metrics.
>
> **W1: Downstream RL requires action labels, limiting the "unsupervised" claim:**
> We wish to clarify the scope of our contributions. The discovery phase (segmentation and hierarchy induction) is fully unsupervised, operating on observations alone; this constitutes the core contribution. In the skill discovery literature (Ranchod et al., 2015; Kipf et al., 2019; Zhu et al., 2022), "skill discovery" refers precisely to identifying reusable behavioural units and their structure from data, not necessarily packaging them as executable options. We go further by demonstrating that these discovered structures have practical utility when deployed as options in downstream RL (Figures 5 and 6).
>
> Regarding action-free deployment: learning executable policies from purely passive observation is generally infeasible without additional assumptions: given only states $s$ and $s'$, one cannot disambiguate which of $N$ possible actions caused a transition. Either online environment interaction (BCO; Torabi et al., 2018) or action labels are necessary for policy learning. Importantly, our segmentation stage is highly data-efficient, achieving strong performance with as few as 100 episodes (Avg. mIoU of 59% from 100 episodes vs. 66% at 500 episodes for Craftax Stone Pickaxe Static), suggesting that correspondingly few action-labelled demonstrations would suffice at deployment.
>
> **W2: The single-skill-per-timestep assumption may be restrictive for robotic domains:**
> We appreciate this point and will expand our limitations discussion in revision. We note that in the domains we target and even in robotic manipulation, non-overlapping skill assumptions remain the standard. NPBRS (Ranchod et al., 2015) was designed for and validated on robotic demonstrations (quadcopter obstacle courses) under this assumption, and ASOT (Xu & Gould, 2024) achieves state-of-the-art on real-world human activity datasets (cooking, assembly) too. Extending to concurrent skills is an interesting direction for future work.
>
> **Minor Points**
> * **Q1 (Inferring K):** Of course, it would be ideal to not have to select $K$. Fortunately, our ablation (Appendix D.1, Table 13) shows graceful degradation when $K$ is overestimated, and HiSD requires only this loose upper bound, whereas CompILE needs the number of segments per trajectory and average skill length, and OMPN needs action labels, ground-truth orderings, and a predefined hierarchy depth. A practical approach to inferring $K$ automatically could be to run HiSD with decreasing values of $K$ starting from a high initial estimate, monitoring statistics such as the frequency of skill-label switching or the number of active clusters; when reducing $K$ causes a sharp increase in switching frequency or segmentation cost, this signals that distinct skills are being merged, providing a natural stopping criterion. This could allow us to get the same effect without requiring non-parametrics as in NPBRS (Ranchod et al., 2015). We will add this as an explicit future direction.
> * **Q2 (Training times):** We apologise for the vagueness in Table 10. Across Craftax tasks, HiSD averages 37 seconds per run versus 1,952 seconds for CompILE (~53x faster). OMPN exhibited similar runtimes to CompILE with higher memory requirements. We will add precise timing data to the revised manuscript.
> * **Q3 (Hyperparameter tuning advantage):** We imposed a uniform 24-hour tuning budget per method. HiSD's computational efficiency is itself a practical contribution, enabling faster iteration. However, even comparing each method's best run, HiSD outperforms on harder tasks. Moreover, both CompILE and OMPN received ground-truth sub-task orderings at inference, which is an advantage HiSD does not use, and yet they still underperformed.
> * **Q4 (Fixed vs. stochastic ordering):** In "static" configurations, every episode follows an identical skill sequence. In "random" configurations, the ordering of skills varies between episodes wherever task dependencies permit (e.g., collecting wood and stone in either order once a wooden pickaxe has been made). This tests whether methods discover consistent skill identities despite variable sequencing.
> * **Q5 (Dataset sizes):** All tasks use 500 expert trajectories. In terms of environment steps, Craftax tasks range from ~7.5k to ~16.5k total steps (e.g., Stone Pickaxe averages ~33 steps/episode; see Table 4, Appendix A.1.1), while Minecraft averages ~786 steps/episode for ~393k total steps (Table 7, Appendix A.2.1). For Minecraft, both the "all" and "mapped" datasets share identical underlying trajectories (same 500 episodes), differing only in labelling granularity: "all" uses 44 fine-grained labels; "mapped" groups semantically similar skills into 14 categories. The full mapping is provided in Table 9 (Appendix A.2).

---

> > ### Author Rebuttal · Reviewer_74Qm · 2026-04-03
> >
> > The authors provided helpful clarifications regarding the experimental setup, training efficiency, and dataset construction. The minor questions (Q1–Q5) have been adequately addressed.
> >
> > However, my main concerns remain only partially addressed. Regarding W1, while the authors argue that action-free deployment is fundamentally infeasible, the original concern was specifically about evaluating settings with fewer action labels, which seems tractable and would better justify the "unsupervised" framing. Regarding W2, the response largely defers the issue to future work without further analysis.
> >
> > Overall, I maintain my positive score of 4 (weak accept).

---

> > > ### Author Response · Authors · 2026-04-06
> > >
> > > We thank the reviewer for their continued engagement and for acknowledging that Q1–Q5 have been adequately addressed.
> > >
> > > W1 (Evaluating with fewer action labels): We appreciate this clarification and agree that the distinction is important. To clarify the terminology: the skill discovery phase (segmentation + hierarchy induction) is fully unsupervised. Here we use the term "skills" as in prior work on skill segmentation (Ranchod et al., 2015; Kipf et al., 2019; Zhu et al., 2022), referring to the identification of reusable behavioural units and their structure from data. However, turning discovered skills into executable options requires learning policies via behavioural cloning, which does need action labels. We agree "unsupervised option discovery" would be an overclaim; our claim is specifically about unsupervised structure discovery.
> > >
> > > That said, the reviewer raises an important practical question: can these discovered structures be deployed with only a small number of action-labelled demonstrations? Motivated directly by this suggestion, we conducted a preliminary experiment on the Craftax Wooden Pickaxe task. We varied the number of action-labelled episodes (N) available for option learning (BC policies, PU classifiers) while keeping the downstream RL evaluation fixed at 100k environment steps. The full sweep covers N ∈ {10, 20, …, 100, 200, 250, 300, 350, 500}. Each entry reports the final mean reward (± 1 SD) averaged over the last 10% of training steps across 3 PPO seeds. Full findings are presented below:
> > >
> > > | N | GT Skills | HiSD Skills | GT Hierarchy | HiSD Hierarchy |
> > > |:---|:---------:|:-----------:|:------------:|:--------------:|
> > > | 10 | 0.00 (0.00) | 0.00 (0.00) | 0.00 (0.00) | 0.00 (0.00) |
> > > | 20 | 0.00 (0.00) | 0.00 (0.00) | 0.00 (0.00) | 0.00 (0.00) |
> > > | 30 | 0.66 (0.47) | 0.00 (0.00) | 0.32 (0.46) | 0.01 (0.01) |
> > > | 40 | 0.96 (0.01) | 0.00 (0.00) | 0.35 (0.28) | 0.00 (0.00) |
> > > | 50 | 0.00 (0.00) | 0.00 (0.00) | 0.00 (0.00) | 0.00 (0.00) |
> > > | 60 | 0.96 (0.04) | 0.65 (0.46) | 1.00 (0.00) | 0.00 (0.00) |
> > > | 70 | 0.96 (0.02) | 0.00 (0.00) | 0.98 (0.02) | 0.00 (0.00) |
> > > | 80 | 0.96 (0.02) | 0.99 (0.00) | 0.98 (0.01) | 0.70 (0.41) |
> > > | 90 | 0.55 (0.41) | 0.39 (0.41) | 0.31 (0.44) | 0.00 (0.00) |
> > > | 100 | 0.00 (0.00) | 0.45 (0.40) | 0.00 (0.00) | 0.31 (0.43) |
> > > | 200 | 0.00 (0.00) | 0.66 (0.47) | 0.67 (0.46) | 0.00 (0.00) |
> > > | 250 | 0.33 (0.47) | 0.67 (0.46) | 0.97 (0.02) | 0.91 (0.11) |
> > > | 300 | 0.65 (0.46) | 0.65 (0.46) | 0.94 (0.05) | 0.99 (0.01) |
> > > | 350 | 0.65 (0.45) | 0.98 (0.01) | 0.98 (0.01) | 0.95 (0.04) |
> > > | 500 | 0.61 (0.43) | 0.97 (0.02) | 1.00 (0.00) | 1.00 (0.00) |
> > >
> > > These results confirm that the discovered hierarchies provide practical value with substantially reduced action-labelled data. The large standard deviations at intermediate N (e.g., 0.46–0.47) reflect bimodal outcomes where runs either converge or fail entirely. Flat skills-only variants show non-zero performance at lower N than their hierarchical counterparts, as expected since composite options compound BC policy errors across stages. Nevertheless, both GT and HiSD hierarchical agents reach near-perfect reward with low variance by N = 300-500, and HiSD Hierarchy already achieves 0.91 (± 0.11) at N = 250, which is relatively low by current RL standards. The segmentation itself remains fully unsupervised and works well with as few as 100 episodes. We will include an analysis in the revised manuscript.
> > >
> > > W2 (Concurrent skills): We would like to offer a small clarification. Our point is not that concurrent skills are unimportant, but rather that the sequential assumption covers a broad and practically significant class of tasks, even in robotic settings. Existing work on real-world activity datasets, including cooking (Xu & Gould, 2024), PC assembly, and quadcopter navigation (Ranchod et al., 2015), all operate successfully under this assumption. We agree that investigating concurrent skill discovery and understanding where it provides downstream advantages over sequential decompositions is an interesting direction, and we will frame this more carefully in the revised limitations and future work section.
> > >
> > > We hope these additional results and clarifications address the reviewer's remaining concerns.

---

### Decision · Program_Chairs · 2026-04-30

**Decision:**

Accept (regular)

**Comment:**

The reviewers found this paper to make a solid contribution to hierarchical skill discovery from offline observational data, with the key strength being that it achieves useful structure learning under weaker assumptions than prior work while evaluating across multiple domains. The empirical results were viewed as technically sound, and the rebuttal was helpful in addressing reviewer questions and clarifying the paper’s scope.

Overall, the discussion supports acceptance. While the level of novelty is moderate, the paper makes a reasonable contribution beyond prior work on similar problems and should be of interest to the community. In the final version, the authors should take the reviewers’ comments into account, particularly in improving clarity and sharpening the presentation of the contribution.